# InP/GaSb core-shell nanowires: A novel hole-based platform with strong spin-orbit coupling for full-shell hybrid devices

Andrea Vezzosi[1,2], Carlos Payá[3], Paweł Wójcik[4], Andrea Bertoni[2],
Guido Goldoni[1,2], Elsa Prada[3] and Samuel D. Escribano[5]★

**1** Dipartimento di Scienze Fisiche, Informatiche e Matematiche,
Università di Modena e Reggio Emilia, Via Campi 213/a, 41125 Modena, Italy
**2** Centro S3, CNR-Istituto Nanoscienze, Via Campi 213/a, 41125 Modena, Italy
**3** Instituto de Ciencia de Materiales de Madrid (ICMM), CSIC, E-28049 Madrid, Spain
**4** AGH University of Krakow, Faculty of Physics and Applied Computer Science,
Al. Mickiewicza 30, 30-059 Krakow, Poland
**5** Department of Condensed Matter Physics, Weizmann Institute of Science,
Rehovot 7610, Israel

★ samuel.diazes@gmail.com

## Abstract

Full-shell hybrid nanowires (NWs), structures comprising a superconductor shell that encapsulates a semiconductor (SM) core, have attracted considerable attention in the search for Majorana zero modes (MZMs). However, the predicted Rashba spin-orbit coupling (SOC) in the SM is too small to achieve substantial topological minigaps. In addition, the SM wavefunction spreads all across the section of the nanowire, leading typically to a finite background of trivial subgap states with which MZMs may coexist. To overcome both problems, we explore the advantages of utilizing core-shell hole-band NWs as the SM part of a full-shell hybrid, with an insulating core and an active SM shell. In particular, we consider InP/GaSb core-shell NWs, which allow to exploit the unique characteristics of the III-V compound SM valence bands. We demonstrate that they exhibit a robust hole SOC that emerges from the combination of the intrinsic spin-orbit interaction of the SM active shell and the confinement effects of the nanostructure, thus depending mainly on SM and geometrical parameters. In other words, the SOC is intrinsic and does not rely on red electric fields, which are non-tunable in a full-shell hybrid geometry. As a result, core-shell SM hole-band NWs are found to be a promising candidate to explore Majorana physics in full-hell hybrid devices, addressing several challenges in the field.

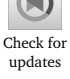

# 1 Introduction

Hybrid superconductor-semiconductor (SC-SM) heterostructures are probably the most analyzed platform [1] for the creation of one-dimensional (1D) topological superconductivity [2,3] and the search of Majorana zero modes [4] (MZMs). These exotic quasiparticles offer immunity to local noise and possess non-trivial braiding statistics [5], making them promising building-blocks of future fault-tolerant quantum computers [2,6,7]. A recently proposed hybrid nanowire (NW) design, the so-called full-shell NW [8], alleviates some of the problems that conventional partial-shell Majorana NWs present [9–13] and that have hindered the creation of MZMs during the last decade [14]. In full-shell NWs, the SM core is fully surrounded by a superconducting metallic shell, shielding the NW from disorder created by the electrostatic environment and surface reconstruction of the NW facets exposed to air. Moreover, the driving mechanism for the topological transition is not a Zeeman field, but an orbital effect associated with the doubly-connected geometry of the parent SC in the presence of a magnetic flux [8,15,16]. Thus, full-shell NWs require much smaller magnetic fields to achieve the topological phase, what prevents the degradation of the parent superconducting state, and operate with small or zero $g$-factor.

    The full-shell architecture, however, also presents some disadvantages [14,17,18]. Most notably, it is not possible to gate tune the chemical potential inside the core due to the metallic encapsulation. Furthermore, using microscopic simulations in realistic Al/InAs full-shell wires, the predicted electron SOC is too small to provide substantial topological protection [19]. In this case, the Rashba SOC appears due to the radial electric field that is produced by the (uncontrollable) SM conduction-band (CB) bending at the SM-SC interface [20–22]. In addition, for the parameter regions for which Majorana states are predicted to appear, these zero modes typically coexist with a number of trivial subgap states that give rise to a dense local density of states (LDOS) background [16]. These subgap states have been dubbed Caroli–de Gennes–Matricon (CdGM) analogs [23], and their presence is due to the doubly-connected geometry of the SC shell and the non-zero winding of the SC pairing phase in the presence of a magnetic flux. In a recent study [16] that thoroughly examines the phenomenology of Al/InAs full-hell hybrid NWs, it has been established that MZMs free from CdGM analogs should be possible in tubular-core NWs, this is, wires where the electron wave function is concentrated in a tube of a certain thickness close to the SM-SC interface.

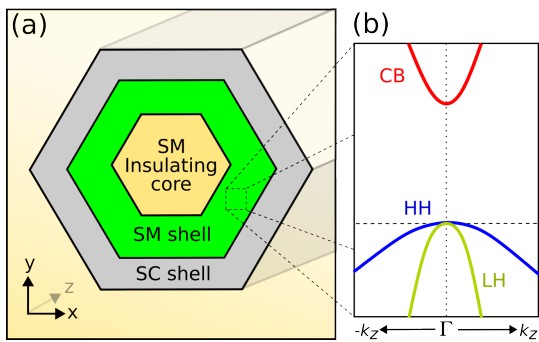

Figure 1: **Sketch of the nanostructure.** (a) Sketch of the proposed full-shell hybrid nanowire (NW). This heterostructure is composed of a semiconductor (SM) core-shell NW made of an insulating core (yellow) and an active SM layer (green). The NW is fully surrounded by a thin superconductor (SC) shell (grey). (b) Typical band structure of a bulk III-V compound SM. For some specific materials, like the InP/GaSb core-shell NWs studied here, the Fermi level (horizontal dashed line) may lie at the valence bands close to the band edge (LH and HH stands for light and heavy-hole bands, and CB for conduction band).

In view of these arguments, in this work we make a specific proposal for the hybrid's core that could potentially improve the performance of full-shell NWs and thus facilitate the creation of MZMs with a large topological gap. We propose to use a core-shell insulator-SM NW, see Fig. 1(a), where the active SM layer is outside, confining the wave function close to the SC-SM interface by means of an insulating core. This should effectively decrease the number of GdGM analogs inside the gap once the SC was included, and enable the formation of topological islands in the phase diagram according to Ref. [16]. Importantly, we make use of the *hole* bands of III-V compound SMs to harness the intrinsic spin-orbit coupled nature of the valence bands (VBs). The SOC of hole bands is typically much larger than that of electrons which consequently may lead to sizeable topological minigaps.

Our proposal centers around a specific material configuration, core-shell InP/GaSb NWs, with GaSb acting as the SM shell and InP as the insulating core, see Fig. 1(a). Although the proposed core-shell structure has not yet been explored experimentally, it seems to be viable.[1] Note that in the proposed device the Fermi level is placed at the VBs, close to the band edge [24, 25], which is convenient for the topological phase when proximitized with the SC. We first examine the SOC of realistic InP/GaSb NWs for different radii and GaSb thicknesses. For this purpose, we use a microscopic approach that employs a self-consistent Schrödinger-Poisson equation within an 8-band k·p Hamiltonian framework. We find substantial values of SOC, of the order of 20 meV·nm, to be compared to values of $2-4$ meV·nm reported in Ref. [19] for similar geometries with InAs. Furthermore, we demonstrate that the SOC does not rely on external factors like electric fields or strain. We then elucidate the origin and characteristics of the SOC and provide analytical expressions derived from an effective Luttinger-Kohn (LK) Hamiltonian model. We find that the SOC in these nanostructures originates from the combination of the intrinsic properties of the SM active layer and the radial confinement, in agreement with similar nanodevices [26, 27].

---

[1]There is a lattice constant mismatch of around 4% between InP and GaSb. While this mismatch is important for the quality of the interface in planar heterostructures, it is not in vapor-liquid-solid NWs. For these NWs, the radial growth allows to accommodate different lattice cells, providing typically a sharp, relaxed interface.

## 2 System and methods

We start by discussing the requirements that a core-shell NW should have to potentially give rise to topologically protected MZMs in a full-shell hybrid geometry. We look for a core-shell heterostructure whose Fermi level lies at the VBs of the shell and close to the VB maximum, see Fig. 1(b). In addition, the core must be insulating at these energies, which depopulates the nanowire center and concentrates the wave function at the outer part of the NW. In this regard, only type-I and type-II SM heterostructures are suitable, whereas type-III heterostructures are not since in these ones the VBs and CBs in different layers overlap. There are several material combinations that could be analyzed, being the lattice mismatch between the two materials a key factor that determines whether the heterostructure is viable.

For the sake of concreteness, we propose to use type-II heterostructures based on Sb compounds because they have hole carriers at the Fermi level more commonly [28]. Particularly, for our numerical simulations we will consider InP/GaSb core-shell NWs. To the best of our knowledge, these NWs have not yet been grown and analyzed experimentally. However, high quality and relaxed GaSb layers grown directly on InP(001) substrates using solid source molecular beam epitaxy have been reported [29, 30]. Furthermore, InAs/InP/GaAsSb core-dual-shell NWs have been recently grown using catalyst-free chemical beam epitaxy [31].

Let us mention that, apart from InP/GaSb, other heterostructures could be contemplated.[2] Related ones, such as GaSb/GaAsSb, should in principle display similar properties. A different option would be to consider group IV SM heterostructures, such as Ge/Si, which is a more popular platform to exploit the SM hole-bands [33]. Ge/Si heterostructures are well-established and widely adopted in SM qubit platforms [34–37] as their VBs exhibit strong SOC [38–41] and electrically tunable $g$-factors [42–44]. However, Ge-based heterostructures face some disadvantages compared to III-V SM compounds. Ge is more prone to dislocations and disorder due to its chemical properties and common growth methods.[3] They also offer less versatility in tailoring band alignment, as III-V materials enable the use of ternary compounds to precisely engineer the electronic properties [61]. In addition, growing a sharp, clean superconducting layer on Ge/Si is also more challenging [45–50]; its surface oxidizes quickly, requiring etching and preparation steps, and exhibits poorer chemical adhesion to SCs. These factors lead to higher disorder in Ge-based heterostructures. And, while this may be less critical for quantum dot physics (in the context where Ge is typically studied), it is crucial for quasi-1D systems like the one analyzed here. Consequently, III-V SM compounds have become more popular for studying SM-SC heterostructures in 1D systems, and particularly in the context of topological superconductivity [62]. Note that several theoretical works have analyzed nanodevices based on these heterostructures for the creation and manipulation of MZMs [63–66], but they have not yet been considered in the context of the full-shell hybrid geometry that we propose here.

Coming back to our proposal, InP/GaSb heterostructures exhibit a type-II band alignment with a wide gap of $\sim 0.5$ eV. Specifically, the VB maximum of GaSb lies within the energy gap of InP, as schematically illustrated in Fig. 2(a). We assume that the Fermi level is placed close to the VB edge of GaSb as reported in Ref. [29]. However, different growing and fabrication

---

[2]A recently published work [32] proposes to use a topological insulator made of BiTe/Se in the place of the SM core in a full-shell hybrid geometry.

[3]For creating Ge/Si-based heterostructures, various epitaxial growing techniques have been employed, including molecular beam epitaxy [53], hybrid epitaxy [54], and low-energy plasma-enhanced chemical vapor deposition [55, 56]. Nonetheless, these methods often require the growth of several microns of material to minimize defect nucleation, leading to elevated surface roughness and a high residual threading dislocation density. An alternative approach using reverse-graded buffers grown via chemical vapor deposition [57] has shown promise for producing smoother, thinner heterostructures with improved mobility [58, 59]. This method however is not yet optimized to consistently yield good quality samples [60], resulting in lower overall electron mobilities compared to III-V compounds.

methods, and particularly the ultimate incorporation of the SC outer shell in the hybrid NW, may change the position of this Fermi level. In this respect, we note that partial control over the wire's overall doping could be achieved through the incorporation of chemical impurities within the InP core during the growth process [29, 67, 68]. Given that the InP core primarily acts as an insulator in this context, it is reasonable to assume that these dopants would exert a minimal influence on the electronic properties of the wire, aside from their impact on the Fermi level position. Note also that other strategies to engineer the band alignment of hybrid quantum devices are currently being investigated. For example, in a recent experimental work [69], argon milling is used to modify the SC-SM interface while maintaining its high quality.

To describe our SM NWs we employ the 8-band k·p Hamiltonian, which accurately reproduces the band structure of III-V compound SMs around the Γ point [70, 71] and thus provides reliable estimations of the SOC [72]. This model is suitable for heterostructures as well, under the assumption that the parameters display an abrupt change at the interface between the SM materials. This is the case of shells epitaxially grown on top of the core when the overall NW doping is low, so that the Fermi wavelength is much larger that the dimensions of the core and shell. We determine the energy spectrum of the core-shell NW through a self-consistent solution of the Schrödinger-Poisson equation (see Appendix A for further details). To achieve this, we employ the finite element method (FEM) on an inhomogeneous grid and implement a Broyden mixing iterative approach [73] using nwkp$_\text{y}$, a Python library recently developed in-house [74]. Our analysis assumes a hexagonal cross-section, consistent with the majority of vapor-liquid-solid III-V compound SM NWs, and considers it is translational invariant along the longitudinal axis ($z$-direction). In this way, we access to the quasi-1D bulk properties of the NW. We make sure that our inhomogeneous FEM grid preserves the $D_6$ symmetry of the hexagonal cross-section, which otherwise could introduce spurious solutions.

# 3 Results

## 3.1 Energy spectrum

A typical energy spectrum of the investigated core-shell NWs is presented in Fig. 2(b), displaying several traverse subbands that arise due to the confinement imposed by the finite wire's cross section. The VB edge is denoted by $E_0$ and it represents the zero-point energy due to the finite size confinement and the mean-field effective potential $\phi(r)$. The color bar shows the light hole (LH) and heavy hole (HH) character of each state. Several noteworthy observations can be made from this spectrum. First, the subband dispersion is negative, as expected for the VBs. Second, all subbands have a hybrid LH and HH character that depends on $k_z$. Notably, the LH character predominates in the higher-energy subbands, i.e. those closer to the Fermi level. Conversely, lower energy subbands have a stronger HH character because they have lower effective mass in the confinement plane. As we shall demonstrate, the hybrid LH and HH character is essential to provide a strong SOC to the different subbands and it is ultimately regulated by the strength of the charge density confinement.

In Fig. 2(c) we present an example of the charge density distribution corresponding to the spectrum computed in Fig. 2(b). It shows the hole concentration throughout the cross section (solid black lines represent the interfaces). The charge density has a ring-like distribution centered approximately at the GaSb shell average radius, with negligible localization within the InP core. The nearly perfect cylindrical symmetry of this charge distribution arises from the finite thickness $w$ of the SM shell and the occupation of the lowest angular momentum states. The $z$-component of the total angular momentum $F_z$ is the sum of the orbital angu-

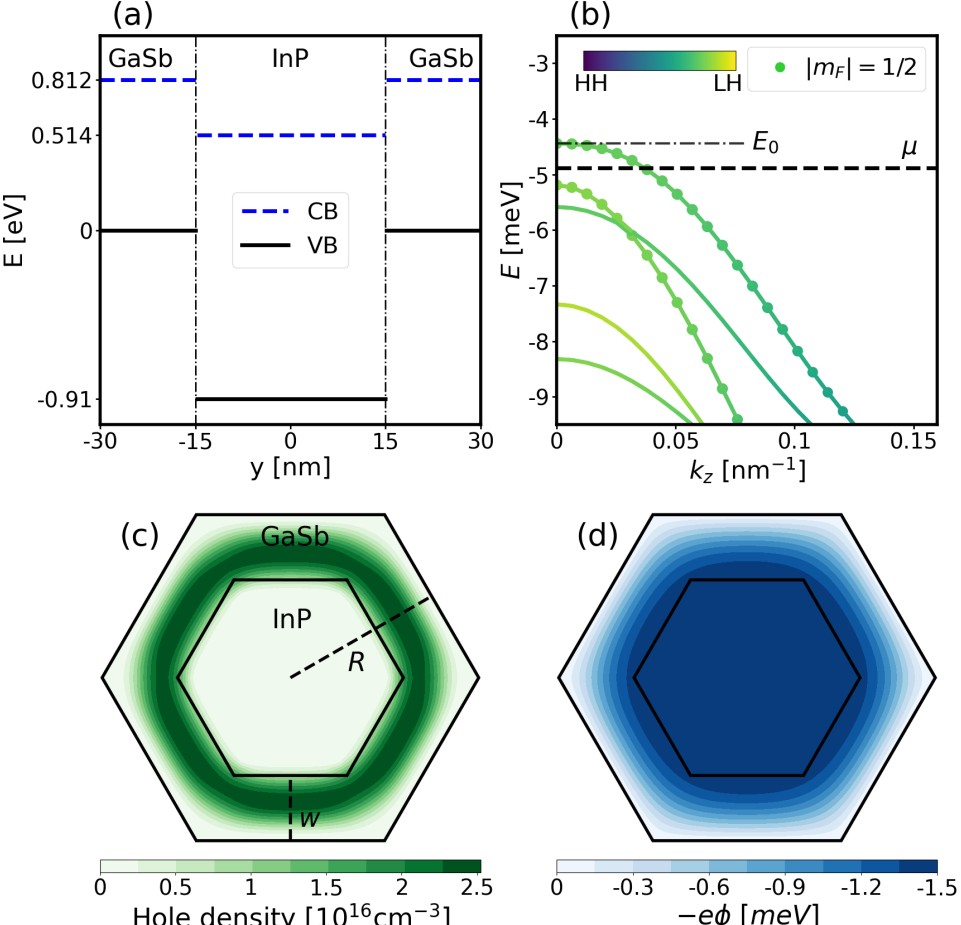

Figure 2: **InP/GaSb core-shell NWs.** (a) Schematic representation of the band alignment of an InP/GaSb heterostructure. This type-II SM heterostructure features the GaSb valence band (VB) maximum positioned within the energy gap of InP. We assume that the Fermi level (black dashed line) lies close to the GaSb VB maximum. (b) Typical energy spectrum of InP/GaSb NWs near the $\Gamma$ point as a function of axial wave vector $k_z$. The colorbar denotes the weight of each state on the light-hole (LH) and heavy-hole (HH) bulk bands. Due to confinement effects, the highest energy subbands exhibit primarily a LH character, albeit with a degree of hybridization between LH and HH states. The projection onto the eigenstates with quantum number $|m_\mathrm{F}| = \frac{1}{2}$ of the total angular momentum is represented by the size of dots at discrete values of $k_z$. $\mu$ is the chemical potential and $E_0$ is the VB edge energy (black dashed-dotted line). (c) Charge density distribution of holes within the core-shell nanowire, corresponding to the spectrum presented in (b). The wire geometry is defined by its radius $R$ and shell thickness $w$. (d) Electrostatic potential energy profile for the same simulation. For these simulations $R = 30$ nm and $w = 15$ nm.

lar momentum $L_z$, with quantum numbers $m_L \in \mathbb{Z}$, and the pseudo-spin $S_z$, with quantum numbers $m_S = \left\{\pm\frac{1}{2}, \pm\frac{3}{2}\right\}$. More details can be found in Appendix A. In a NW with cylindrical symmetry and within the axial approximation, $F_z$ commutes with the Hamiltonian and thus $m_F = m_L + m_S$ is a good quantum number, which takes the values $m_F \in (\mathbb{Z} + 1/2)$. Subbands in this limit come in pairs separated by a finite energy gap and with the same $m_F$. These subbands consist of nearly opposite spin states with dominant orbital angular momentum $m_L$ and $m_L + 1$. At zero magnetic field the states corresponding to $m_F$ and $-m_F$ are doubly degenerate. For simplicity, we focus on those with $m_F > 0$.

In Fig. 2(b) we calculate the projection of the different states onto the $m_F = \frac{1}{2}$ value, represented by the size of the dots at discrete values of $k_z$. The subbands with $m_F = \frac{1}{2}$ (or $m_F = -\frac{1}{2}$) are of importance as they can give rise to MZMs in full-shell hybrid geometries [19]. It is noteworthy that the two higher energy subbands in Fig. 2(b) predominantly correspond to the $m_F = \frac{1}{2}$ sector, meaning that a cylindrical approximation is justified for them. The cylindrical approximation remains valid in general to treat the eigenstates of the hexagonal wire at low dopings. Only when the shell thickness $w$ is rather small, the charge distribution tends to develop an hexagonal shape due to the hybridization with higher angular harmonics $m_F + 6n$ for integer $n$. Notice that the rest of the visible subbands in Fig. 2(b) have essentially zero projection onto $m_F = \frac{1}{2}$ since they correspond predominantly to other, larger $m_F$ values. To find another couple of predominantly $m_F = \frac{1}{2}$ subbands, one would need to populate the higher radial subbands. Here, nevertheless, we are considering core-shell wires with a small shell thickness $w$, meaning that there is a large energy splitting between radial subbands and thus one would need to strongly increase the chemical potential or doping of the core-shell wire to populate them. Since we are interested in tubular-core wires with the smallest possible number of CdGM analog states, we concentrate only on the first $m_F = \frac{1}{2}$ subband pair corresponding to the lowest-energy radial subband. In addition, this will allow us to provide analytical approximations for the SOC and other parameters.

To complement this discussion, we present the electrostatic potential energy profile $-e\phi(r)$ for the same parameters in Fig. 2(d). It is computed self-consistently taking into account the potential generated by the hole charges within the nanowire. We set the boundary condition as $\phi_b = 0$ at the wire facets, assuming that there is a surrounding metal that is grounded. Consequently, the potential energy is zero in proximity to the interface and becomes progressively negative towards the core. This implies that holes tend to feel attracted towards the outer interface. Note, however, that the hole wave function average radius is slightly shifted towards the core in Fig. 2(c). The reason is the different boundary conditions between the outer and inner interfaces.[4] It is important to note that if the metallic shell is superconducting, $\phi_b \neq 0$ even if it is grounded. This is due to the Ohmic SC-SM contact that produces a SM band bending at the interface as a result of the work-function difference between both materials. The precise magnitude and shape of this band-bending depends on chemical details that have not yet been studied for this heterostructure. In general, it is reasonable to assume that it will create a negative potential energy at the outer interface, pushing holes away from it and towards the inner interface. This might be very convenient for our full-shell hybrid NW proposal, as this might decrease metallization effects from the SC [75,76] while still preserving the necessary proximity effect due to the finite thickness of the SM layer.

---

[4]The envelope-function equations are solved considering an infinite barrier for the outer surface of the NW (so that the wave function is set to zero there), while the inner barrier is penetrable corresponding to a SM-insulator interface. In Fig. 2(c) the inner barrier has a height of around 0.9 eV [see Fig. 2(a)], whereas the electrostatic energy is of the order of a few meV [see Fig. 2(c)], not enough to counteract the confinement energy.

## 3.2 Spin-orbit coupling

We now proceed to study the SOC in the considered core-shell SM NWs. Using the energy spectrum and wave functions, we calculate numerically both the SOC $\alpha$ and the effective mass by fitting each subband pair close to $k_z = 0$ with a standard dispersion relation (see Appendix A for the rather elaborate method). We anticipate that in Sec. 3.3 we will derive an analytical Hamiltonian with the same dispersion. Generically, the SOC refers to the strength of the coupling between the momentum of a quasiparticle to its pseudo-spin. While the pseudo-spin in traditional cases refers to the electron's pure spin (e.g., the conduction-band spin), in our work, it refers to a more complex band combination of spin-like degrees of freedom (see Appendix B for the specific band-combination involved).

In Fig. 3(a), we present $\alpha$ of the highest-energy subband pair, i.e., the one closest to the band edge, as a function of the SM shell thickness $w$ for different NW radii $R$, each represented by a different color. We extract the SOC when the chemical potential is located halfway between the two $m_F = \frac{1}{2}$ subbands [see Fig. 2(b)]. We consider realistic values for $R$ and $w$, ranging from $R = 15$ nm to $R = 40$ nm, and from $w = 5$ nm to $w = R - 10$ nm.[5] Larger radii are certainly possible experimentally, but the methodology that we employ to extract $\alpha$ ceases to be valid for $R \gtrsim 50$ nm. For these core-shell NWs, we obtain SOCs in the range of 15 to 30 meV·nm. These numbers are roughly a factor of $\sim 4$ larger than the SOCs found in Ref. [19] for solid-core NWs based on CB electrons and equivalent radius $R$. As in the case of Ref. [19], we find that $\alpha$ increases as $R$ decreases. In our case of a tubular core, the SOC moreover increases with $w$ for a fixed $R$.

Concerning the SOC axis, the dominant component must be radial in a NW with a (nearly) cylindrical symmetry as ours, since it is the only direction with a broken inversion symmetry. One could naively think that the SOC should vanish due to the centrosymmetry of the cylindrical geometry. However, notice that $\vec{\alpha} = \alpha \hat{r}$ itself is not an observable, and we should rather consider the spin-orbit field $\vec{\Omega} \sim \vec{\alpha} \times \vec{\sigma}$. While the expectation value $\langle \vec{\alpha} \rangle$ is strictly zero in a problem with cylindrical symmetry, $\langle \vec{\Omega} \rangle$ is not since $\vec{\sigma}$ depends on position for the eigenstates. Intuitively, this can be understood by realizing that both the SOC and the spin change sign at opposite cross section points, so that they contribute constructively to the average.

In Fig. 3(b) we show the behavior of the SOC with the doping level of the wire, considering different shell thicknesses $w$ for a fixed radius $R = 30$ nm. Notably, $\alpha$ remains almost constant with $\mu$, what suggests that the hole bands are essentially insensitive to the electric field inside the wire. This is in sharp contrast to what happens to the CB of these materials, where the sole contribution to the (Rashba) SOC arises from the electric field [70].

Both observations that, for a fixed $R$, $\alpha$ increases with $w$ and is essentially independent of the electric field, point to an intrinsic nature of the hole-band SOC of III-V compound SM NWs. To further understand this behavior and unravel the underlying origin of the SOC, in Fig. 3(d) we conduct a comparative analysis by selectively omitting certain contributions in the multiband Hamiltonian. In blue we present the results derived from the complete 8-band model, which are the same as the blue curve in Fig. 3(a). First, we test whether the interaction between the HH-LH bands of GaSb and either the split-off bands or the CBs of both GaSb or InP, do influence the SOC. To explore this, we set either the split-off gaps $\Delta_\beta$ to exceedingly large values or the coupling with the CBs $P_\beta$ to zero. We recompute the energy spectrum and the SOC under each assumption, and depict the results in Fig. 3(d) with different colors. Notice that none of these couplings contribute significantly to $\alpha$ as all the curves lie almost on top of each other.

---

[5]Regarding the values of $w$ that we consider, the minimum thickness of 5 nm ensures that the SM shell can be grown with sufficient homogeneity, while we take $w = R - 10$ nm for the maximum thickness since the minimum core radius that is typically grown nowadays is approximately 10 nm.

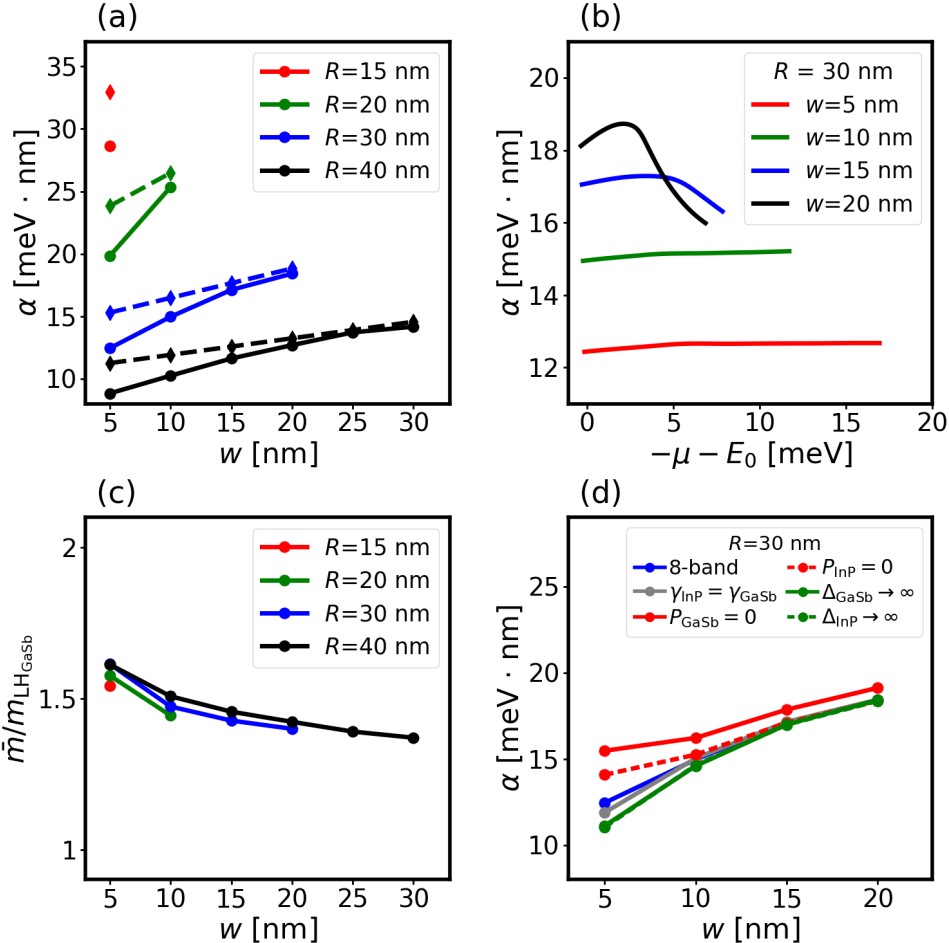

Figure 3: **SOC of InP/GaSb core-shell NWs.** (a) SOC $\alpha$ of the highest-energy subband pair as a function of GaSb shell thickness $w$ for different wire radii $R$, each represented by a different color. Solid lines correspond to 8-band model numerical results while dashed lines to results obtained analytically with Eq. (3). $\alpha$ decreases with $R$ and increases with $w$. (b) $\alpha$ vs chemical potential of the wire ($-\mu$) with respect to the VB edge $E_0$, for $R = 30$ nm and various $w$. The negative sign of $-\mu$ stems from the occupation of the VB. Notably, there is a weak dependence on doping and, consequently, on the electric field inside the nanostructure. (c) Harmonic mean effective mass $\bar{m}$ [see Eq. (B.20)] of the same subband pair as in (a) normalized to the light-hole bulk effective mass of GaSb $m_{\mathrm{LH}_{\mathrm{GaSb}}} = 0.0439 m_0$, being $m_0$ the free electron mass. $\bar{m}$ does not change substantially with $w$. (d) $\alpha$ vs $w$ for $R = 30$ nm, considering various scenarios: infinite split-off gaps for GaSb (solid green) or InP (dashed green), zero coupling between CBs and VBs for GaSb (solid red) or InP (dashed red), and identical Kane parameters (except for the gaps) for InP and for GaSb (grey). Results computed with the full 8-band model (blue) are also shown for comparison. The different curves lie almost on top of each other. All these results suggest an intrinsic origin of the SOC of the GaSb hole bands.

Next, we consider whether the SOC arises from the inversion asymmetry generated by the core-shell interface [77]. Specifically, we investigate whether the non-commutativity of the Kane parameters with momentum at the interface plays an important role. To test this, we set all Kane parameters (excluding the gaps) of InP to be identical to those of GaSb. Remarkably, the results depicted in grey in Fig. 3(d) reveal that the interfacial effects do not have a significant influence on the SOC of the GaSb hole bands.

These results lead us to conclude that the origin of the SOC is an inherent property of the GaSb shell. Specifically, it is due to the radial confinement imposed by the nanostructure which breaks the translation symmetry of the crystal structure of this tetravalent SM. Furthermore, as InP plays a negligible role, we can assume that the core serves as a generic wide band-gap insulator in our simulations, and that could be replaced by other insulating material. Nevertheless, the particular chemical properties of the core are still relevant to determine the Fermi level within the hole bands of GaSb.

### 3.3 Effective Hamiltonian and analytical results

Given the previous observations, it is justified to describe the NW through a Luttinger-Kohn (LK) Hamiltonian that only involves the two spinful hole bands (the LH and the HH) of GaSb. In Appendix B we show that, starting from the LK Hamiltonian, it is possible to derive a $2 \times 2$ effective Hamiltonian that describes the two highest-energy $m_F = \frac{1}{2}$ subbands, given by

$$H_{\text{eff}} = \left( E_{\text{mean}} + \frac{\hbar^2 k_z^2}{2\bar{m}} \right) \sigma_0 + \left( \frac{\delta E}{2} + \frac{\hbar^2 k_z^2}{2m_\delta} \right) \sigma_z + \alpha_{\text{eff}} k_z \sigma_y \,. \tag{1}$$

Here, $\sigma_i$ are Pauli matrices for the effective $\pm 1/2$ spin degree of freedom, $E_{\text{mean}}$ is the mean energy of the two subbands and $\delta E$ the subband splitting (and thus $E_0 = E_{\text{mean}} + \delta E/2$), $\bar{m}$ and $m_\delta$ are the harmonic mean of the effective mass and its dispersion, respectively, and $\alpha_{\text{eff}}$ is the effective SOC. The derivation of this Hamiltonian and the analytical expressions of the different parameters are shown in the Appendix B. Particularly, for the effective mass and the SOC we have

$$\bar{m} = \frac{2m_1 m_2}{m_1 + m_2} \,, \tag{2}$$

$$\alpha_{\text{eff}} = \frac{\gamma_s}{m_0} \hbar^2 \sqrt{3} \chi \left( \frac{1}{R} + \frac{w}{2R^2} \right) \,, \tag{3}$$

where $m_{1,2}$ is the effective mass of each subband [see Eq. (B.21)]. In Eq. (3), $\chi$ is an overlap factor that depends on the degree of LH-HH hybridization of the two subbands, $\gamma_s = (\gamma_2 + \gamma_3)/2$ with $\gamma_i$ the Luttinger parameters of GaSb, and $m_0$ is the free electron mass.

To asses the validity of the effective Hamiltonian of Eq. (1), we evaluate $\alpha_{\text{eff}}$ using the analytical expression of Eq. (3) and the tabulated values for $\gamma_s$ and $\chi$ [78] (see Table 1). We show the results in Fig. 3(a) with dashed lines for comparison with the numerical results. Even though the agreement is not perfect, it is fairly good. This is specially remarkable in light of the various approximations made to derive Eq. (1). This attests to the overall validity of the analytical approach.

The analytical derivation of this effective Hamiltonian helps to clarify the origin of the effective SOC, which turns out to be the same as the one predicted in Refs. [26, 27] for Ge-based heterostructures. Although the bulk SM possesses intrinsic spin-orbit interaction, this interaction does not cause spin-splitting within the HH or LH bands themselves (i.e., no splitting between spin-up and spin-down states within a single band). Instead, it manifests as a splitting of the valence bands into HH and LH bands due to their different pseudo-spin quantum numbers ($m_S = \pm\frac{3}{2}$ for HH and $m_S = \pm\frac{1}{2}$ for LH). But in a nanostructure, confinement

breaks the translational symmetry of the bulk crystal, leading to a quantization of momentum in the confined directions ($r$ and $\varphi$). This quantization causes mixing between the HH and LH bands, making them no longer distinguishable by their pseudo-spin projections $m_S$. Instead, only the total angular momentum $m_F$ is conserved. As a result, the subbands, which are now hybridized mixtures of HH and LH states, can exhibit a $k$-dependent spin splitting, similar to a conventional spin-orbit interaction.

Notice that, in accordance with our numerical 8-band model simulations, the effective SOC of Eq. (3) is independent of the electric field, decreases with $R$, and increases with $w$. Moreover, $\alpha_{\text{eff}}$ also increases with the degree of LH-HH hybridization $\chi$, which is ultimately regulated by the degree of wave function confinement. This supports again that the origin of the SOC is not an electric field, as it happens for the CB, but rather an orbital effect imposed by the wave function confinement. In a cylinder, $\alpha_{\text{eff}}$ thus points in the radial direction,[6] along which the spatial symmetry is broken. We emphasize that a strong radial component of the SOC, perpendicular to an applied axial magnetic field, is crucial to obtain a topological superconducting phase with sizeable topological minigaps [8, 16].

### 3.4 Topological chemical-potential window

To further characterize the probability of hosting a topological superconducting phase in a full-shell geometry, it is convenient to study the energy splitting of the subband pair with total angular momentum $m_F = \frac{1}{2}$. As mentioned before, these are the only ones capable of hosting a topological phase (in the absence of mode mixing perturbations [16]). The $m_F = \frac{1}{2}$ subbands appear in pairs, with one pair corresponding to each radial subband. To find a topological phase, the chemical potential $\mu$ must be placed between two topological subbands (with an odd number of filled topological subbands due to the even-odd effect [15]). For sufficiently small $R$ and/or $w$, the different $m_F = \frac{1}{2}$ subband pairs do not overlap and hence the topological phase diagram is composed of non-overlapping topological regions for each radial mode [16]. The separation between consecutive subbands within a pair thus defines the parameter window for which a topological superconducting phase is possible and, consequently, the likelihood of encountering MZMs at a given chemical potential or doping.

In Fig. 4(a) we depict the value $-\mu$ for which the chemical potential crosses the highest-energy $m_F = \frac{1}{2}$ subband pair vs the shell width $w$ for different wire radii $R$. Solid lines indicate the crossing with the first subband of the pair, while dashed lines indicate the crossing with the second one. In Fig. 4(b) we explicitly display the topological window $\Delta E_{\text{TS}}$, which is the energy spacing between solid and dashed lines of Fig. 4(a). For all the cases analyzed in Fig. 4(a), the topological windows are larger than typical superconducting gaps, e.g., $\sim 0.2$ meV for Al or $\sim 0.5$ meV for Sn. Notably, the topological window increases with decreasing $R$ and increasing $w$, just as happened with the SOC magnitude [see Fig. 3(a)]. This can be readily understood by examining the analytical equations governing these pairs. For a fixed $R$, thicker shells result in smaller values of the average wave function radius, $R_{\text{av}}$, that lead to a greater level spacing in turn. This agrees with the analytical expression for $\delta E$ in Eq. (B.13), which is proportional to $R_{\text{av}}^{-2} = (R - \frac{w}{2})^{-2}$, as shown in Ref. [27].

Our analysis demonstrates that thicker shells yield stronger SOC and wider topological windows. Conversely, smaller diameters also result in enhanced SOC and increased spacing between radial subbands. We clarify that the ultimate topological window is also determined by the induced superconducting gap when the outer SC shell is included in the nanostructure. If the superconducting gap is smaller than the chemical potential window $\Delta E_{\text{TS}}$, then it pro-

---

[6]Note that the Hamiltonian of Eq. (1) is written in a rotated basis independent of the angle $\varphi$. When the rotation is unfolded, the spin-orbit interaction term becomes $\alpha_{\text{eff}} k_z \sigma_\varphi$ (along with other terms), which clearly indicates that $\alpha_{\text{eff}}$ must be radial. This is because the vectors $\vec{\alpha}$, $\vec{k}$ and $\vec{\sigma}$ must all be mutually perpendicular.

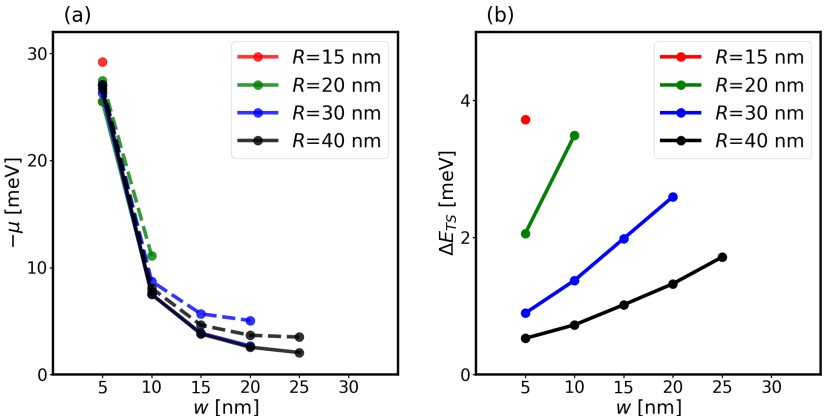

Figure 4: **Topological chemical-potential window.** (a) Crossing points of the chemical potential $-\mu$ with the first (solid lines) and the second (dashed lines) highest-energy $|m_{\mathrm{F}}| = \frac{1}{2}$ subbands vs shell thickness $w$ for different NW radii $R$ (in different colors). The spacing between the two lines defines the energy window $\Delta E_{\mathrm{TS}}$ for which a topological SC phase is possible, which is plotted in panel (b).

vides a cutoff for the topological window size. Hence, the larger the parent superconducting gap is (and the stronger the SC-SM coupling), the larger the topological window up to the value fixed by $\Delta E_{\mathrm{TS}}$.

## 4 Discussion and conclusions

In this work we have made a proposal to enhance the performance of full-shell hybrid NWs. In the device configurations studied so far [8], the NWs should suffer from tiny SOCs [19], which weaken the topological protection, and should be plagued with CdGM subgap analog states [16, 23] that spoil the potential MZMs, at least in the absence of fairly specific mode-mixing perturbations [8, 15, 16]. We have suggested to use a tubular-shaped SM core made of a core-shell III-V compound SM [see Fig. 1(a)], being the SM core an insulator and the SM shell a $p$-doped layer. On the one hand, the tubular shape reduces the spread of the wave function across the NW section, confining it to the region close to the SC-SM interface. According to what happens in electron-based full-shell hybrid NWs, this should dramatically reduce the number of CdGM analogs coexisting with the MZMs [16]. On the other hand, the $p$-doped SM shell allows to harness the intrinsic nature of the SOC of hole bands.

Specifically, we have considered core-shell InP/GaSb NWs, whose Fermi level lies at the VB edge of the GaSb shell. Our investigations unveiled that core-shell InP/GaSb NWs exhibit a robust SOC, with values of $\sim 20$ meV·nm for NW radius of $\sim 30$ nm, that does not depend on electric fields or strain at the interface with the SC or the insulating core. Instead, it depends on the degree of HH and LH hybridization, ultimately regulated by the confinement strength provided by the NW radius. The intrinsic nature of the SOC thus eliminates additional device constrains to enhance the SOC, like tuning an electric field inside the NW or using ultrathin NWs.

Using microscopic self-consistent numerical simulations based on the 8-band model, we observed that the SOC in our core-shell NWs increases with decreasing NW radius $R$, just as happens with CB NWs, meaning that smaller cross-section wires will potentially develop larger topological minigaps. This behavior could be explained by the reduction of the confinement, responsible of the intrinsic SOC, as $R$ increases [70]. Nonetheless, smaller radii require larger magnetic fields to achieve the topological phase, which may be detrimental to the parent su-

perconductor. Therefore, a trade-off $R$ must be chosen based on the SC material.[7] Additionally, the SOC increases with shell thickness $w$. The reason is that increasing $w$ reduces the energy separation between LH and HH bands. Consequently the LH-HH coupling increases and thus the SOC. We found a similar trend with $R$ and $w$ for the topological window of the core-shell NW with chemical potential in the absence of other subgap states. However, since increasing $w$ too much should also increase the number of CdGM analogs in the presence of the SC shell [16], a sweet spot for $w$ must also be found for optimal topological protection.

The core-shell NW behavior with $R$ and $w$ was corroborated with analytical calculations derived from the LK Hamiltonian. We derived a simple and practical expression for the effective SOC in Eq. (3), which produced remarkably similar results to the numerical calculations. Analytical approximations for other NW parameters, such as effective masses and subband energy splittings, were also derived in Appendix B.

The insulating core in our core-shell wires, apart from decreasing the presence of trivial states within the gap, could additionally mitigate the impact of disorder stemming from impurities introduced during the NW growth process. This is simply because the wave function is constrained to be located within the reduced cross-section of the active SM tube, instead of spreading throughout the whole NW area, thereby reducing the exposure to impurities. This argument is valid both for electron-based and hole-based tubular-core NWs. Apart from this, we expect disorder along the NW's axis to be as detrimental for MZM physics as it is in conventional Majorana NWs, although it has been recently shown [66] that disorder effects have a smaller impact in the hole bands.

Finally, let us mention that in this work we have explored the potential of hole-based core-shell NWs for the topological performance of full-shell hybrid NWs based on previous conclusions derived with CB SMs [8, 16, 19, 23], but we have not conducted simulations in the presence of an actual SC and a magnetic flux. Concerning the later, it is known that in full-shell hybrid NWs, the orbital effect of the magnetic field is crucial for the development of the topological phase [8, 16]. In these geometries, subgap states cannot be tuned through gate potentials, because the SC encapsulation shields external electric fields inside the SM core; and barely tuned by the Zeeman effect, because Zeeman splittings are typically weak for small magnetic fields. Instead, subgap states are strongly tuned by an *effective* Zeeman field given in terms of the the magnetic flux [8]. Importantly, if the superconducting shell is smaller than the London penetration length, the magnetic flux in the SM evolves continuously with the applied field, leading to a topological phase whose flux extension is determined by the effective radius of the SM, among other parameters [16]. This is an advantage compared to other systems with a thick SC shell [81], which experience a quantization of the magnetic flux inside the SM, imposing stricter conditions for a topological phase. While orbital effects are critical for Majorana physics, a detailed analysis of their impact in our nanostructure is beyond the scope of this work, particularly given the complex spin and orbital structure of hole bands. Nonetheless, confinement-induced splitting between LH and HH bands suggests that orbital effects in the higher-energy subbands should behave similarly to those in conventional CB SMs.

Regarding the inclusion of the SC shell, we would like to point out that the specific details of the superconducting proximity effect on a VB SM remain unclear and are the subject of current research. Some initial works [82, 83] suggest that there could be important differences with respect to the CB counterpart, although a strong enough superconducting proximity effect should be possible [47, 48, 50]. This aspect nevertheless demands a thorough analysis that necessarily needs the experimental input and is also beyond the scope of this work.

---

[7]Notice that the magnetic field required to drive the full-shell hybrid system into the topological phase depends on the choice of the SC material, as the Little-Parks effect [79, 80] depends on the SC's critical temperature and diffusive coherence length. For instance, with Al as the SC shell, we estimate that a magnetic field of approximately 0.2 T would be sufficient to reach the first possible topological region for a NW with $R = 40$ nm, whereas a field of 1.7 T would be required for a NW with $R = 15$ nm.

Concerning specific materials for the SC, the most common in conventional full-shell NWs is Al. Admittedly, achieving epitaxial growth of Al on GaSb may pose challenges due to the significant lattice mismatch between the two materials. Alternatively, one could explore SCs like Sn [84,85] or Nb [86,87], which have been grown on other III-V compound SMs and posses similar lattice constants as GaSb. The epitaxial growth ensures a clean, sharp SM-SC interface, which ultimately provides a hard induced gap. However, despite the epitaxial growth, the SC can exhibit cross hatched patterns or spatially dependent interfaces, which is detrimental for Majorana physics, not only for full-shell but for partial-shell hybrid nanowires as well. Remarkably, some degree of disorder, either in the SC or at the SM/SC interface, is actually beneficial as it breaks parallel momentum conservation, allowing for a better hybridization between the SC and the SM [88].

Metallization effects from the SC into the SM can be important in hybrid nanowires.[8] Since metallization effects could be specially important in tubular-core NWs (due to the enhanced proximity of the SM wave function to the SC-SM interface), a convenient strategy could be depositing a thin layer of an insulating material between the SC and the SM [91]. This approach helps to control the coupling between the SC and the SM, reducing the effects of metallization and, additionally, opening the possibility to tune the hybrid structure into the topological phase [16]. In particular, the insulating layer could be ternary SM, such as InGaSb, between GaSb and the SC, which could further assimilate the potentially different lattice constants and, thus, expand the range of possible SCs. We believe that the main conclusions of our work should not be affected by the particular SC as long as these arguments are taken into account.

On a final note, we should mention that we do not expect full-shell hybrid NWs based on our SM core-shell nanostructures to behave better against disorder than conventional solid-core full-shell hybrid NWs, except perhaps due to the finite wave function spread across the NW section thanks to the insulating core. Regarding smooth parameter inhomogeneities at the end of the hybrid NW and the formation of quasi-Majoranas with which true MZMs could be mistaken, again, we do not expect advantages in our design (the tubular core), since those effects depend on variations along the length of the wire (not across its section). Also, we do not anticipate new experimental signatures specific to the core-shell structure of the NW. All the well-established methods from previous studies, such as detecting zero-bias peaks [8] and Coulomb blockade spacing [92] in conductance experiments, remain applicable to our nanodevices. The true advantages of full-shell hybrid NWs (as compared to partial-shell ones) are their shielding from external electrostatic disorder thanks to the all-around SC shell, the need for much smaller magnetic fields to achieve the topological phase, and the expectation of MZMs at specific (and thus recognizable) flux intervals in certain regions of parameter space. The potential advantages of the type of SM core-shell NWs we propose and charactarize here are the enhancement of the SOC, and the concentration of the SM wave function in a tubular region close to the SC-SM interface. We have argued that, under certain conditions, both properties will facilitate the formation of MZMs and increase their topological minigaps once the SC and orbital effects are included.

---

[8]If the coupling between the SM and SC is too strong, the SC can metallize the SM, diminishing the SOC of the heterostructure among other NW parameters [75,89,90]. This is a known problem of hybrid NWs in general. In addition, in full-shell NWs the coupling to the SC may modify the average radius of the wavefunction inside the SM, effectively increasing the NW radius $R$. This in turn would decrease the effective SOC in our core-shell NWs according to Eq. (3). Nonetheless, since the SC shell is typically very thin (below 10 nm), we assume that this would be a small correction.

# Acknowledgments

We thank Pablo San-Jose for insightful discussions on the numerical implementation. We also thank Hadas Shtrikman, Filip Křížek and Francesco Rossella for fruitful discussions on NW materials and growth processes.

**Funding information**
This research was supported by Grants PID2021-122769NB-I00, PID2021-125343NB-I00 and PRE2022-101362 funded by MICIU/AEI/10.13039/501100011033, "ERDF A way of making Europe" and "ESF+", European Union's Horizon Europe research and innovation program under the Marie Skłodowska-Curie grant agreement number 101120240, and the program "Excellence initiative – research university" for the AGH University.

# A  8-band model for the core-shell nanowire

We consider a semiconductor (SM) hexagonal core-shell nanowire (NW) with its axis along the $z$-coordinate and its section across the $(x, y)$ plane. To obtain its band structure we use the 8-band k·p model [93] with the envelope-function approximation [94]. This is the approximation used to extend the bulk k·p theory to nanostructures by writing the wave function as a linear combination of (lattice periodic) Bloch basis functions, so-called Kane basis functions, with spatially varying coefficients, i.e., the envelope functions. The wave function at axial wave vector $k_z$ is then written as $|\Psi(\mathbf{r}, k_z)\rangle = \sum_{\nu=1}^{8} e^{ik_z z}\psi_\nu(x, y)|\nu\rangle$, being $|\nu\rangle = |S, S_z\rangle$ an eigenstate of the Bloch total pseudo-spin angular momentum $\mathbf{S}$, and $S_z$ the Bloch pseudo-spin angular momentum about the $z$-axis. We choose this axis along the (111) crystallographic direction, which coincides with the growth axis of the NW. In particular, the Kane basis functions are given by

$$|\nu\rangle \in \left\{ \left|\frac{1}{2}, \frac{1}{2}\right\rangle_{\text{EL}}, \left|\frac{1}{2}, -\frac{1}{2}\right\rangle_{\text{EL}}, \left|\frac{3}{2}, \frac{3}{2}\right\rangle_{\text{HH}}, \left|\frac{3}{2}, -\frac{3}{2}\right\rangle_{\text{HH}}, \right.$$
$$\left. \left|\frac{3}{2}, \frac{1}{2}\right\rangle_{\text{LH}}, \left|\frac{3}{2}, -\frac{1}{2}\right\rangle_{\text{LH}}, \left|\frac{1}{2}, \frac{1}{2}\right\rangle_{\text{SO}}, \left|\frac{1}{2}, -\frac{1}{2}\right\rangle_{\text{SO}} \right\}, \quad \text{(A.1)}$$

where the label EL refers to the $s$-like conduction band (CB) Bloch orbitals, while the labels HH, LH and SO correspond to the $p$-like valence band (VB) Bloch orbitals, also referred to as heavy, light and split-off holes, respectively. The energy subbands $E(k_z)$ and the in-plane envelope-function components $\psi_\nu(x, y)$ are obtained by a self-consistent numerical solution of the coupled envelope-function equations, $(H_{8B})_{\mu\nu}\psi_\nu = E\psi_\mu$, and the two-dimensional Poisson equation $\vec{\nabla}\cdot\left(\epsilon(x, y)\vec{\nabla}\phi(x, y)\right) = -\rho(x, y)$, where $\phi(x, y)$ is the electrostatic potential inside the NW, $\rho(x, y)$ is the charge density corresponding to the occupied VB states, and $\epsilon(x, y) = \epsilon_r(x, y)\epsilon_0$ is the dielectric constant.

Details about the specific form of the 8-band Hamiltonian $H_{8B}$ and the self-consistent procedure employed to derive it can be found in Ref. [74]. The parameters of the 8-band model we use for the simulations of the main text are reported in Table 1. Here, the modified Luttinger parameters are calculated as [95]

$$\tilde{\gamma}_1 = \gamma_1 - \frac{E_P}{3\Delta_g}, \quad \tilde{\gamma}_2 = \gamma_2 - \frac{E_P}{6\Delta_g}, \quad \tilde{\gamma}_3 = \gamma_3 - \frac{E_P}{6\Delta_g}, \quad \text{(A.2)}$$

where $\gamma_i$ are material-dependent parameters parameterizing the mixture of HHs and LHs at the top of the VB of cubic SMs, $E_P$ is the Kane energy and $\Delta_g$ is the SM bulk gap. As suggested

Table 1: Material parameters used in the 8-band k·p model at temperature $T = 0$ K. The bulk energy gap $\Delta_g$, the CB and VB offsets $\Delta E_c$ and $\Delta E_v$ at the InP/GaSb interface, the split-off energy $\Delta_{so}$, the Kane energy $E_P$, the conduction-band effective mass $m_e$ and the bare Luttinger parameters $\gamma_i$ are taken from Ref. [78]. $E_P^{mod}$ is the modified Kane energy Eq. (A.3) and $\tilde{\gamma}_i$ are the modified values Eq. (A.2). $\epsilon_r$ refers to the relative dielectric constant of each material.

|  | InP | Interface | GaSb |
|---|---|---|---|
| $\Delta_g$ [eV] | 1.4236 |  | 0.812 |
| $\Delta E_c$ [eV] |  | 0.514 |  |
| $\Delta E_v$ [eV] |  | 0.91 |  |
| $\Delta_{so}$ [eV] | 0.108 |  | 0.76 |
| $E_P$ / $E_P^{mod}$ [eV] | 20.7 / 18.3 |  | 27 / 24.8 |
| $m_e$ [$m_0$] | 0.0795 |  | 0.039 |
| $\gamma_1$ / $\tilde{\gamma}_1$ | 5.08 / 0.79 |  | 13.4 / 3.2 |
| $\gamma_2$ / $\tilde{\gamma}_2$ | 1.60 / $-$0.55 |  | 4.7 / $-$0.39 |
| $\gamma_3$ / $\tilde{\gamma}_3$ | 2.10 / $-$0.047 |  | 6.0 / 0.9 |
| $\epsilon_r$ | 11.77 |  | 15.7 |

by Foreman in Ref. [96], in order to avoid spurious solutions in the 8-band k·p theory, we take a modified definition for the Kane energy parameter, i.e.,

$$E_P^{mod} = \frac{\Delta_g(\Delta_g + \Delta_{so})}{\Delta_g + \frac{2}{3}\Delta_{so}}\left(\frac{m_0}{m_e}\right), \tag{A.3}$$

where $m_0$ is the free electron mass, $m_e$ is the CB effective electron mass and $\Delta_{so}$ is the split-off gap of the bulk SM. Note that the modified Luttinger parameters $\tilde{\gamma}_i$ are calculated from Eqs. (A.2) with $E_P = E_P^{mod}$.

We solve the envelope-function equations using the finite element method (FEM). The real space representation of the Hamiltonian $H_{8B}$ is done on a two-dimensional hexagonal domain, partitioned in a $D_6$ symmetry-compliant mesh of triangular elements. The use of a discretization that is compliant with the symmetry of the confinement potential is fundamental in order to reproduce the expected degeneracies without the use of an an extremely high number of grid points. To reduce the computational burden of the numerical diagonalization, we map the envelope-function equations on inhomogeneous domain partitions, with higher density of points in the conductive GaSb shell with respect to the insulating InP core region. In addition, to avoid the emergence of discretization-related spurious solutions, we solve the envelope-function equations using a mixed polynomial basis, where third-order Hermite polynomials are used for the $s$-like CB components and second-order Lagrange polynomials are used for for the $p$-like VB components [97].

We now define the total angular momentum operator as $F_z = L_z + S_z$, where $L_z$ is the $z$-component of the orbital angular momentum. As explained in Sec. 2 of the main text, $m_F$ are good quantum numbers of $F_z$ for a NW with cylindrical symmetry. It thus characterizes the different traverse subbands in such a limit. The $m_F = \frac{1}{2}$ ($m_F = -\frac{1}{2}$) subband is special because, in the presence of superconductivity in a full-shell hybrid geometry, it can give rise to MZMs in the $n = -1$ ($n = 1$) Little-Parks lobe [8]. We focus for simplicity in $m_F = \frac{1}{2}$ because in the absence of magnetic field, both $m_F = \pm\frac{1}{2}$ subbands are degenerate. Coming back to the hexagonal NW, we now want to obtain the projections of the different 8-band-model states onto the value $m_F = \frac{1}{2}$, since we want to understand which NW subbands are susceptible of becoming topological in the presence of superconductivity. We employ the following procedure

to that end. First, we consider the following eigenvalue equation

$$F_z |\Phi\rangle = m_F^* |\Phi\rangle \,, \tag{A.4}$$

where $|\Phi(\mathbf{r}, k_z)\rangle = \sum_{\nu=1}^8 f_\nu(x, y) |S, S_z\rangle e^{ik_z z}$ and $m_F^*$ are hexagonal-NW eigenstates and eigenvalues of the total angular momentum, respectively. Note that in the position representation, $L_z = -i\hbar(x\partial_y - y\partial_x)$. We then solve Eq. (A.4) using the FEM on the hexagonal domain and obtain a set of $n_{\text{dof}} \times 8$ eigenvalues end eigenvectors, being $n_{\text{dof}}$ the total number of vertices in the mesh. Since the equation is solved on an hexagonal domain, thus breaking cylindrical symmetry, we expect a continuous distribution of the eigenvalues of $F_z$ along the real axis. However, we find that the density of eigenvalues is tightly peaked around half-integer values $\mathbb{Z} + 1/2$. This indicates that the wave function distributes with a quasi-cylindrical symmetry. To quantify the projections of each NW state onto a single $m_F$ target sector, we evaluate the following quantity for each value of the wave-vector $k_z$,

$$C_{m_F} = \frac{1}{N} \sum_{\pm|m_F^*|\in I_{m_F}} \left| \left\langle \Phi_{m_F^*} \middle| \Psi \right\rangle \right|^2 \,, \tag{A.5}$$

where the sum is performed considering the set of eigenvalues $m_F^*$ that fall within a given interval $I_{m_F}$ centered around $m_F$. Notice $N = \sum_{m_F^*} \left| \left\langle \Phi_{m_F^*} \middle| \Psi \right\rangle \right|^2$ is a normalization factor required as the two set of eigenstates are not necessarily orthonormal to each other. In particular, in Fig. 2(b) of the main text we have used this procedure for the target $m_F = \frac{1}{2}$.

We now describe how we extract the values of the phenomenological spin-orbit coupling (SOC) coefficient $\alpha$ and effective mass of the highest-energy subband pair of the hexagonal NW, see Fig. 3. We employ a method that is similar to the one used in Ref. [19]. We assume that these subbands are well described by the following $2 \times 2$ fitting Hamiltonian

$$H_{\text{fit}}(k_z) = \left( E_{\text{mean}} + \frac{\hbar^2 k_z^2}{2\bar{m}} \right) \sigma_o + \left( \frac{\delta E}{2} + \frac{\hbar^2 k_z^2}{2m_\delta} \right) \sigma_z + \alpha k_z \sigma_y \,, \tag{A.6}$$

where $E_{\text{mean}}$ is the mean energy of the two subbands at $k_z = 0$, $\delta E$ is the energy splitting at $k_z = 0$ due to the difference in orbital angular momentum between the two states, and $\bar{m}$ is the harmonic mean of the effective masses $m_1$ and $m_2$ of the two subbands, i.e.,

$$\bar{m} = \frac{2m_1 m_2}{m_1 + m_2} \,, \tag{A.7}$$

and its dispersion is

$$\frac{1}{m_\delta} = \frac{m_2 - m_1}{2m_1 m_2} \,. \tag{A.8}$$

Here, $\sigma_i$ are Pauli matrices for an *effective* spin-1/2 degree of freedom. Note that, in contrast to the case of spin-1/2 electrons in SM nanowires, the highest-energy hole-subband pair has contributions from both states with spin projection $\pm 3/2$ (HHs) and $\pm 1/2$ (LHs). We will show this explicitly, and how Eq. (A.6) can be derived from the Luttinger-Kohn Hamiltonian, in the next Appendix. The energy spectrum of $H_{\text{fit}}$ is

$$E_\pm^{\text{fit}}(k_z) = \left( E_{\text{mean}} + \frac{\hbar^2 k_z^2}{2\bar{m}} \right) \pm \left( \frac{\delta\widetilde{E}(k_z)}{2} \right) \sqrt{1 + \frac{4\alpha^2 k_z^2}{\delta\widetilde{E}(k_z)^2}} \,, \tag{A.9}$$

where

$$\delta\widetilde{E}(k_z) = \delta E + \frac{\hbar^2 k_z^2}{m_\delta} \,. \tag{A.10}$$

In principle, one could use this equation to fit the results of the 8-band model and extract the coefficients $\alpha$, $\bar{m}$ and $m_\delta$. But the effect of the SOC is dominated by the energy splitting $\delta E$ and thus a fitting cannot resolve it. To overcome this issue we follow the procedure proposed in Ref. [19]. First, we diagonalize the 8-band k·p Hamiltonian $H_{8B}$ at $k_z = 0$ and find the $q$ lowest-energy eigenvectors $X_{n_{dof}\times q}$. The matrix $X$ diagonalizes the Hamiltonian $H_{8B}$, such that $X^\dagger H_{8B} X = \Lambda$, where $\Lambda$ is a $q \times q$ matrix containing the eigenvalues $\lambda_i$ $(i = 1,...,q)$ on the diagonal, and $X^\dagger O X = \mathbb{1}_{q\times q}$, being $O$ a $n_{dof} \times n_{dof}$ real and symmetric matrix which is present due to the non-orthogonal basis set used in the FEM. Now we construct the $q \times q$ matrix $D = \text{diag}(-\frac{\delta E}{2}, +\frac{\delta E}{2}, 0, ..., 0)$ for each Kramers pair. The matrix $D$ is then used to shift the two subbands of the first subband pair (for each Kramers pair) so that the two states have the same energy at $k_z = 0$. To implement the energy shifting technique, we compute the matrix $\delta H_{8B} = O X D X^\dagger O^\dagger$ and we diagonalize the Hamiltonian $H_{8B}^{shift}(k_z) = H_{8B}(k_z) + \delta H_{8B}$ as a function of $k_z$. In the perturbed spectrum $\delta E \simeq 0$ and the spin-orbit effect is now pronounced. Although we included the parameter $m_\delta$ in the fitting procedure, to keep into account that the two subbands can have different effective masses, in Fig. 3 of the main text we show results taking into account only the harmonic mean effective mass $\bar{m}$ of the subband pair (as in Ref. [19]). The reason is that $1/m_\delta$ is in general very small and the extracted values or $\alpha$ are essentially independent of $m_\delta$.

## B    Analytical approximations for the core-shell nanowire

The low-energy physics of hole-based nanostructures can be accurately described through a four-band Luttinger-Kohn (LK) Hamiltonian [70]. The isotropic LK Hamiltonian in 3D can be written as

$$H_{LK} = \left(\gamma_1 + \frac{5}{2}\gamma_s\right)\frac{\hbar^2 k^2}{2m_0} - \frac{\hbar^2 \gamma_s}{m_0}(\mathbf{k}\cdot\mathbf{S})^2 \, , \tag{B.1}$$

where $m_0$ is the free electron mass, $\gamma_i$ are the Luttinger parameters specified in Appendix A, $\gamma_s \equiv (\gamma_2 + \gamma_3)/2$, $\hbar\mathbf{k} = -i\hbar\nabla$ is the vector of momentum operators (with $k^2 = -\nabla^2$), and $\mathbf{S}$ is the vector of Bloch pseudospin-3/2 operators. This Hamiltonian is written in the basis $\left\{\left|S_z = +\frac{3}{2}\right\rangle, \left|S_z = +\frac{1}{2}\right\rangle, \left|S_z = -\frac{1}{2}\right\rangle, \left|S_z = -\frac{3}{2}\right\rangle\right\}$, which is the LH-HH subspace of the 8-band model basis of Eq. (A.1). Since $(\gamma_3 - \gamma_2)/\gamma_1 \ll 1$ in GaSb, anisotropic corrections to the LK Hamiltonian are small and the isotropic approximation is well justified.

As we justify in the main text, the physics of an InP/GaSb core-shell is generally well described with a cylindrical approximation. Hence, we use cylindrical coordinates $(r, \varphi, z)$ for the Hamiltonian, with $\mathbf{k} = (k_r, k_\varphi, k_z)$ transforming as $\mathbf{k} \rightarrow \left(-i\left(\partial_r + 1/2r\right), -i\frac{1}{r}\partial_\varphi, -i\partial_z\right)$. Notice that with this definition of the radial momentum we ensure the hermiticity of the operator [98]. Also, since we assume that the NW is translationally invariant along the NW's axis, $k_z$ is a good quantum number and there is no need to replace it by its derivative. Moreover, as the total angular momentum $F_z = \hbar k_\varphi + \hbar S_z$ commutes with the Hamiltonian, one can remove the angular dependence by rotating the LK Hamiltonian through the unitary operator $U = e^{i(m_F - S_z)\varphi}$, being $m_F \in \mathbb{Z} + \frac{1}{2}$ the quantum numbers of $F_z$. This rotation acts on the relevant operators as

$$U^\dagger \mathbf{k} U = \mathbf{k} - \mathbf{e}_\varphi(S_z - m_F), \tag{B.2}$$

$$U^\dagger \mathbf{S} U = \mathbf{e}_r S_x + \mathbf{e}_\varphi S_y + \mathbf{e}_z S_z, \tag{B.3}$$

$$U^\dagger k^2 U = k_r^2 - \frac{1}{4r^2} + \frac{(S_z - m_F)^2}{r^2} + k_z^2. \tag{B.4}$$

In these expressions, we have used the commutation relations $\left[k_r, r^{-1}\right] = ir^{-2}$, $\left[k_\varphi, \mathbf{e}_\varphi\right] = i\mathbf{e}_r$ and $\left[k_\varphi, \mathbf{e}_r\right] = -\frac{i}{r}\mathbf{e}_\varphi$.[9] We can now write down the rotated Hamiltonian $\tilde{H}_{\mathrm{LK}} = U^\dagger H_{\mathrm{LK}} U$ using the standard representation of the $4 \times 4$ angular momentum matrices $\mathbf{S}$ for spin-3/2 operators. The matrix elements of this Hamiltonian are [99]

$$
\begin{aligned}
\left(\tilde{H}_{\mathrm{LK}}\right)_{11} &= \frac{\hbar^2}{2m_0}\left\{(\gamma_1 + \gamma_s)\left[k_r^2 - \frac{1}{4r^2} + \frac{1}{r^2}\left(m_{\mathrm{F}} - \frac{3}{2}\right)^2\right] + (\gamma_1 - 2\gamma_s)k_z^2\right\}, \\
\left(\tilde{H}_{\mathrm{LK}}\right)_{22} &= \frac{\hbar^2}{2m_0}\left\{(\gamma_1 - \gamma_s)\left[k_r^2 - \frac{1}{4r^2} + \frac{1}{r^2}\left(m_{\mathrm{F}} - \frac{1}{2}\right)^2\right] + (\gamma_1 + 2\gamma_s)k_z^2\right\}, \\
\left(\tilde{H}_{\mathrm{LK}}\right)_{33} &= \frac{\hbar^2}{2m_0}\left\{(\gamma_1 - \gamma_s)\left[k_r^2 - \frac{1}{4r^2} + \frac{1}{r^2}\left(m_{\mathrm{F}} + \frac{1}{2}\right)^2\right] + (\gamma_1 + 2\gamma_s)k_z^2\right\}, \\
\left(\tilde{H}_{\mathrm{LK}}\right)_{44} &= \frac{\hbar^2}{2m_0}\left\{(\gamma_1 + \gamma_s)\left[k_r^2 - \frac{1}{4r^2} + \frac{1}{r^2}\left(m_{\mathrm{F}} + \frac{3}{2}\right)^2\right] + (\gamma_1 - 2\gamma_s)k_z^2\right\}, \\
\left(\tilde{H}_{\mathrm{LK}}\right)_{12} &= -\frac{\gamma_s}{m_0}\hbar^2\sqrt{3}\,k_z\left[k_r + \frac{i}{2r} - \frac{i}{r}\left(m_{\mathrm{F}} - \frac{1}{2}\right)\right], \\
\left(\tilde{H}_{\mathrm{LK}}\right)_{13} &= -\frac{\gamma_s}{m_0}\frac{\sqrt{3}}{2}\hbar^2\left[k_r^2 - \frac{1}{4r^2} - \frac{2i}{r}\left(m_{\mathrm{F}} - \frac{1}{2}\right)\left(k_r + \frac{i}{2r}\right) - \frac{1}{r^2}\left(m_{\mathrm{F}} + \frac{1}{2}\right)\left(m_{\mathrm{F}} - \frac{3}{2}\right)\right], \\
\left(\tilde{H}_{\mathrm{LK}}\right)_{14} &= \left(\tilde{H}_{\mathrm{LK}}\right)_{23} = 0, \\
\left(\tilde{H}_{\mathrm{LK}}\right)_{24} &= -\frac{\gamma_s}{m_0}\frac{\sqrt{3}}{2}\hbar^2\left[k_r^2 - \frac{1}{4r^2} - \frac{2i}{r}\left(m_{\mathrm{F}} + \frac{1}{2}\right)\left(k_r + \frac{i}{2r}\right) - \frac{1}{r^2}\left(m_{\mathrm{F}} - \frac{1}{2}\right)\left(m_{\mathrm{F}} + \frac{3}{2}\right)\right], \\
\left(\tilde{H}_{\mathrm{LK}}\right)_{34} &= \frac{\gamma_s}{m_0}\hbar^2\sqrt{3}\,k_z\left[k_r + \frac{i}{2r} - \frac{i}{r}\left(m_{\mathrm{F}} + \frac{3}{2}\right)\right].
\end{aligned}
\tag{B.5}
$$

The rest of the elements can be derived taking into account that $\tilde{H}_{\mathrm{LK}}$ must be hermitian.

Most of the topological properties of the hybrid NW are set from the properties of this Hamiltonian close to $k_z = 0$. Hence, we will use perturbation theory to obtain a manageable analytical expression that applies for small $k_z$. To this end, we split the Hamiltonian in two parts

$$
\widetilde{H}_{\mathrm{LK}} = H_0 + H', \tag{B.6}
$$

where $H_0 \equiv \widetilde{H}_{\mathrm{LK}}(k_z{=}0)$ is treated as the zeroth-order Hamiltonian, and $H' \equiv \widetilde{H}_{\mathrm{LK}}(k_z) - H_0$ as the perturbative term. The zeroth-order eigenstates $|i, m_{\mathrm{F}}\rangle$ of $H_0$ with energy $E_{i,m_{\mathrm{F}}}$ are in general a linear combination

$$
|i, m_{\mathrm{F}}\rangle = \sum_\beta f_{m_{\mathrm{F}}, i, \beta}(r)|\beta\rangle |m_{\mathrm{F}}\rangle, \tag{B.7}
$$

where $i = \{1, 2, 3, 4\}$ labels the four possible bands and $f_{m_{\mathrm{F}}, i, \beta}$ are (envelope-function) coefficients that mix the four bulk bands $\beta = \left\{\mathrm{HH}_\uparrow, \mathrm{LH}_\uparrow, \mathrm{LH}_\downarrow, \mathrm{HH}_\downarrow\right\}$ for each $m_{\mathrm{F}}$. Notice that the LK Hamiltonian $H_0$ is separated in two disconnected blocks at $k_z = 0$: one couples the states $|S_z = +3/2\rangle$ with $|S_z = -1/2\rangle$, and the other couples the states $|S_z = -3/2\rangle$ with $|S_z = +1/2\rangle$. As a result, one can explicitly write the linear combinations that provide the four eigensates

$$
\begin{aligned}
|1, m_{\mathrm{F}}\rangle &= \left(f_{m_{\mathrm{F}}, 1, \mathrm{HH}_\uparrow}(r)\left|\mathrm{HH}_\uparrow\right\rangle + f_{m_{\mathrm{F}}, 1, \mathrm{LH}_\downarrow}(r)\left|\mathrm{LH}_\downarrow\right\rangle\right)|m_{\mathrm{F}}\rangle, \\
|2, m_{\mathrm{F}}\rangle &= \left(f_{m_{\mathrm{F}}, 2, \mathrm{LH}_\uparrow}(r)\left|\mathrm{LH}_\uparrow\right\rangle + f_{m_{\mathrm{F}}, 2, \mathrm{HH}_\downarrow}(r)\left|\mathrm{HH}_\downarrow\right\rangle\right)|m_{\mathrm{F}}\rangle, \\
|3, m_{\mathrm{F}}\rangle &= \left(f_{m_{\mathrm{F}}, 3, \mathrm{HH}_\uparrow}(r)\left|\mathrm{HH}_\uparrow\right\rangle + f_{m_{\mathrm{F}}, 3, \mathrm{LH}_\downarrow}(r)\left|\mathrm{LH}_\downarrow\right\rangle\right)|m_{\mathrm{F}}\rangle, \\
|4, m_{\mathrm{F}}\rangle &= \left(f_{m_{\mathrm{F}}, 4, \mathrm{LH}_\uparrow}(r)\left|\mathrm{LH}_\uparrow\right\rangle + f_{m_{\mathrm{F}}, 4, \mathrm{HH}_\downarrow}(r)\left|\mathrm{HH}_\downarrow\right\rangle\right)|m_{\mathrm{F}}\rangle.
\end{aligned}
\tag{B.8}
$$

---

[9]We recall that in cylindrical coordinates $\partial_\varphi \mathbf{e}_\varphi = -\mathbf{e}_r$ and $\partial_\varphi \mathbf{e}_r = \mathbf{e}_\varphi$.

The corresponding Kramers partners $|i, -m_F\rangle$ can be obtained by applying the time reversal operator $\Theta = e^{-i\pi S_y/\hbar} K$ on the eigenstates in Eq. (B.8), being $S_y$ a standard spin-3/2 matrix and $K$ the complex conjugate operator.

The radial confinement present in our core-shell NW allows us to make further simplifications. First of all, we can write the envelope functions $f_{m_F,i,\beta}(r)$ as a linear combination of orthonormal wave functions $\psi_{m_r}(r)$ that represent radial subbands $m_r$, i.e., $f_{m_F,i,\beta}(r) = \sum_{m_r} c^{(m_r)}_{m_F,i,\beta} \psi_{m_r}(r)$. For strong confinement, different $m_r$ are (roughly) decoupled (i.e., they are not mixed at finite $k_z$) and we can focus on the lowest-energy radial subband $m_r = 1$. Thus, we write

$$f_{m_F,i,\beta}(r) = c_{m_F,i,\beta} \psi_1(r), \tag{B.9}$$

with the normalization condition

$$\sum_\beta \left| c_{m_F,i,\beta} \right|^2 = 1, \quad \forall m_F, i. \tag{B.10}$$

Notice that we have removed the label $m_r = 1$ from the expressions for simplicity. Secondly, the confinement will make those bands with larger weight on the LH sector to remain higher in energy as they posses a lower kinetic energy [see kinetic terms in Eq. (B.5)]. Moreover, due to the orthonormalization conditions of the eigenstates $\langle i, m_F | j, m_F \rangle = \delta_{ij}$, we have

$$\left| c_{m_F,3,HH_\uparrow} \right| = \left| c_{m_F,1,LH_\downarrow} \right|, \quad \text{and} \quad \left| c_{m_F,3,LH_\downarrow} \right| = \left| c_{m_F,1,HH_\uparrow} \right|. \tag{B.11}$$

This means that the weights on the HH and LH sectors of $i = 3$ are exchanged with respect to the ones of $i = 1$ (up to a phase). The same happens for $i = 2$ and $i = 4$. Hence, without loss of generality we choose $|1, m_F\rangle$ and $|2, m_F\rangle$ as the states with the largest weights on the LH bands, and disregard $|3, m_F\rangle$ and $|4, m_F\rangle$ since the LH-HH splitting for $m_r = 1$ is larger than than the splitting between $m_r$ and $m_{r+1}$. Particularly, the LH-HH energy splitting is of the order of $\frac{2\gamma_s \hbar^2 \pi^2}{m_0 w^2} \sim 80 \times \left(\frac{10\,\text{nm}}{w}\right)^2$ meV, while the radial subband splitting goes as $\frac{3\hbar^2 \pi^2 \gamma_1}{2m_0 w^2}\left(1 - \frac{2\gamma_s}{\gamma_1}\right) \sim 30 \times \left(\frac{10\,\text{nm}}{w}\right)^2$ meV [27], where $w$ is the SM shell thickness.

We thus obtain a $2 \times 2$ effective Hamiltonian by performing first-order perturbation theory onto the subspace spanned by the two states $\{|1, m_F\rangle, |2, m_F\rangle\}$ for the first radial subband. The matrix elements $\langle i, m_F | H_0 | j, m_F \rangle$ in this basis yield the eigenenergies $E_{m_F,1}$ and $E_{m_F,2}$ of the two subbands at the $\Gamma$ point. The matrix elements of $H'$ have to be computed as

$$\langle i, m_F | H' | j, m_F \rangle = \int_0^R dr \sum_{\beta,\beta'}^4 f^*_{m_F,i,\beta} H'_{\beta\beta'} f_{m_F,j,\beta'}. \tag{B.12}$$

After some algebra, the effective Hamiltonian can be written in a compact form as

$$H_{\text{eff},m_F} \simeq \left( E_{\text{mean},m_F} + \frac{\hbar^2 k_z^2}{2\bar{m}_{m_F}} \right)\sigma_0 + \left( \frac{\delta E_{m_F}}{2} + \frac{\hbar^2 k_z^2}{2m_{\delta,m_F}} \right)\sigma_z + \alpha_{\text{eff},m_F}\sigma_y k_z, \tag{B.13}$$

with Pauli matrices for the subband degree of freedom. Note that Eq. (B.13) is analogous to the Hamiltonian in Eq. (A.6) for $m_F = \frac{1}{2}$, but we now can find functional expressions for the parameters. Particularly, $E_{\text{mean},m_F}$ and $\delta E_{m_F}$ correspond to the mean energy and energy splitting at $k_z = 0$ between the two subbands of the $m_F$ pair, respectively, and they are given

by

$$
\begin{aligned}
E_{\text{mean},m_{\text{F}}} &= \frac{E_{m_{\text{F}},1} + E_{m_{\text{F}},2}}{2} \\
&= \frac{\hbar^2}{2m_0} \frac{(\gamma_1 + \gamma_s)}{2} \left( \chi_{\text{HH}_\uparrow,\text{HH}_\uparrow}^{m_{\text{F}},1,1} + \chi_{\text{HH}_\downarrow,\text{HH}_\downarrow}^{m_{\text{F}},2,2} \right) \mathcal{I}_1 \\
&\quad + \frac{\hbar^2}{2m_0} \frac{(\gamma_1 + \gamma_s)}{2} \left\{ \left[ \left( m_{\text{F}} - \frac{3}{2} \right)^2 - \frac{1}{4} \right] \chi_{\text{HH}_\uparrow,\text{HH}_\uparrow}^{m_{\text{F}},1,1} + \left[ \left( m_{\text{F}} + \frac{3}{2} \right)^2 - \frac{1}{4} \right] \chi_{\text{HH}_\downarrow,\text{HH}_\downarrow}^{m_{\text{F}},2,2} \right\} \mathcal{I}_2 \\
&\quad + \frac{\hbar^2}{2m_0} \frac{(\gamma_1 - \gamma_s)}{2} \left( \chi_{\text{LH}_\downarrow,\text{LH}_\downarrow}^{m_{\text{F}},1,1} + \chi_{\text{LH}_\uparrow,\text{LH}_\uparrow}^{m_{\text{F}},2,2} \right) \mathcal{I}_1 \\
&\quad + \frac{\hbar^2}{2m_0} \frac{(\gamma_1 - \gamma_s)}{2} \left\{ \left[ \left( m_{\text{F}} + \frac{1}{2} \right)^2 - \frac{1}{4} \right] \chi_{\text{LH}_\downarrow,\text{LH}_\downarrow}^{m_{\text{F}},1,1} + \left[ \left( m_{\text{F}} - \frac{1}{2} \right)^2 - \frac{1}{4} \right] \chi_{\text{LH}_\uparrow,\text{LH}_\uparrow}^{m_{\text{F}},2,2} \right\} \mathcal{I}_2 \\
&\quad - \frac{\gamma_s}{m_0} \frac{\sqrt{3}}{2} \hbar^2 \frac{1}{2} \left( 2\,\text{Re}\left\{ \chi_{\text{HH}_\uparrow,\text{LH}_\downarrow}^{m_{\text{F}},1,1} \right\} + 2\,\text{Re}\left\{ \chi_{\text{LH}_\uparrow,\text{HH}_\downarrow}^{m_{\text{F}},2,2} \right\} \right) \mathcal{I}_1 \\
&\quad + \frac{\gamma_s}{m_0} \frac{\sqrt{3}}{2} \hbar^2 \frac{1}{2} \left[ \left( m_{\text{F}} + \frac{1}{2} \right) \left( m_{\text{F}} - \frac{3}{2} \right) + \frac{1}{4} \right] 2\,\text{Re}\left\{ \chi_{\text{HH}_\uparrow,\text{LH}_\downarrow}^{m_{\text{F}},1,1} \right\} \mathcal{I}_2 \\
&\quad + \frac{\gamma_s}{m_0} \frac{\sqrt{3}}{2} \hbar^2 \frac{1}{2} \left[ \left( m_{\text{F}} - \frac{1}{2} \right) \left( m_{\text{F}} + \frac{3}{2} \right) + \frac{1}{4} \right] 2\,\text{Re}\left\{ \chi_{\text{LH}_\uparrow,\text{HH}_\downarrow}^{m_{\text{F}},2,2} \right\} \mathcal{I}_2 \\
&\quad + \frac{\gamma_s}{m_0} \frac{\sqrt{3}}{2} \hbar^2 \frac{1}{2} 2i \left( m_{\text{F}} - \frac{1}{2} \right) 2\,\text{Im}\left\{ \chi_{\text{HH}_\uparrow,\text{LH}_\downarrow}^{m_{\text{F}},1,1} \right\} \mathcal{I}_3 \\
&\quad + \frac{\gamma_s}{m_0} \frac{\sqrt{3}}{2} \hbar^2 \frac{1}{2} 2i \left( m_{\text{F}} + \frac{1}{2} \right) 2\,\text{Im}\left\{ \chi_{\text{LH}_\uparrow,\text{HH}_\downarrow}^{m_{\text{F}},2,2} \right\} \mathcal{I}_3 ,
\end{aligned}
\tag{B.14}
$$

$$
\begin{aligned}
\frac{\delta E_{m_{\text{F}}}}{2} &= \frac{E_{m_{\text{F}},1} - E_{m_{\text{F}},2}}{2} \\
&= \frac{\hbar^2}{2m_0} \frac{(\gamma_1 + \gamma_s)}{2} \left( \chi_{\text{HH}_\uparrow,\text{HH}_\uparrow}^{m_{\text{F}},1,1} - \chi_{\text{HH}_\downarrow,\text{HH}_\downarrow}^{m_{\text{F}},2,2} \right) \mathcal{I}_1 \\
&\quad + \frac{\hbar^2}{2m_0} \frac{(\gamma_1 + \gamma_s)}{2} \left\{ \left[ \left( m_{\text{F}} - \frac{3}{2} \right)^2 - \frac{1}{4} \right] \chi_{\text{HH}_\uparrow,\text{HH}_\uparrow}^{m_{\text{F}},1,1} - \left[ \left( m_{\text{F}} + \frac{3}{2} \right)^2 - \frac{1}{4} \right] \chi_{\text{HH}_\downarrow,\text{HH}_\downarrow}^{m_{\text{F}},2,2} \right\} \mathcal{I}_2 \\
&\quad + \frac{\hbar^2}{2m_0} \frac{(\gamma_1 - \gamma_s)}{2} \left( \chi_{\text{LH}_\downarrow,\text{LH}_\downarrow}^{m_{\text{F}},1,1} - \chi_{\text{LH}_\uparrow,\text{LH}_\uparrow}^{m_{\text{F}},2,2} \right) \mathcal{I}_1 \\
&\quad + \frac{\hbar^2}{2m_0} \frac{(\gamma_1 - \gamma_s)}{2} \left\{ \left[ \left( m_{\text{F}} + \frac{1}{2} \right)^2 - \frac{1}{4} \right] \chi_{\text{LH}_\downarrow,\text{LH}_\downarrow}^{m_{\text{F}},1,1} - \left[ \left( m_{\text{F}} - \frac{1}{2} \right)^2 - \frac{1}{4} \right] \chi_{\text{LH}_\uparrow,\text{LH}_\uparrow}^{m_{\text{F}},2,2} \right\} \mathcal{I}_2 \\
&\quad - \frac{\gamma_s}{m_0} \frac{\sqrt{3}}{2} \hbar^2 \frac{1}{2} \left( 2\,\text{Re}\left\{ \chi_{\text{HH}_\uparrow,\text{LH}_\downarrow}^{m_{\text{F}},1,1} \right\} - 2\,\text{Re}\left\{ \chi_{\text{LH}_\uparrow,\text{HH}_\downarrow}^{m_{\text{F}},2,2} \right\} \right) \mathcal{I}_1 \\
&\quad + \frac{\gamma_s}{m_0} \frac{\sqrt{3}}{2} \hbar^2 \frac{1}{2} \left[ \left( m_{\text{F}} + \frac{1}{2} \right) \left( m_{\text{F}} - \frac{3}{2} \right) + \frac{1}{4} \right] 2\,\text{Re}\left\{ \chi_{\text{HH}_\uparrow,\text{LH}_\downarrow}^{m_{\text{F}},1,1} \right\} \mathcal{I}_2 \\
&\quad - \frac{\gamma_s}{m_0} \frac{\sqrt{3}}{2} \hbar^2 \frac{1}{2} \left[ \left( m_{\text{F}} - \frac{1}{2} \right) \left( m_{\text{F}} + \frac{3}{2} \right) + \frac{1}{4} \right] 2\,\text{Re}\left\{ \chi_{\text{LH}_\uparrow,\text{HH}_\downarrow}^{m_{\text{F}},2,2} \right\} \mathcal{I}_2 \\
&\quad + \frac{\gamma_s}{m_0} \frac{\sqrt{3}}{2} \hbar^2 \frac{1}{2} 2i \left( m_{\text{F}} - \frac{1}{2} \right) 2\,\text{Im}\left\{ \chi_{\text{HH}_\uparrow,\text{LH}_\downarrow}^{m_{\text{F}},1,1} \right\} \mathcal{I}_3 \\
&\quad - \frac{\gamma_s}{m_0} \frac{\sqrt{3}}{2} \hbar^2 \frac{1}{2} 2i \left( m_{\text{F}} + \frac{1}{2} \right) 2\,\text{Im}\left\{ \chi_{\text{LH}_\uparrow,\text{HH}_\downarrow}^{m_{\text{F}},2,2} \right\} \mathcal{I}_3 ,
\end{aligned}
\tag{B.15}
$$

where we have defined the overlaps

$$
\chi_{\beta,\beta'}^{m_{\text{F}},i,j} \equiv c_{m_{\text{F}},i,\beta}^* c_{m_{\text{F}},j,\beta'} ,
\tag{B.16}
$$

and the radial integrals

$$\mathcal{I}_1 \equiv \int_0^R \mathrm{d}r\, r\, \psi_1^*(r) k_r^2 \psi_1(r), \tag{B.17}$$

$$\mathcal{I}_2 \equiv \int_0^R \mathrm{d}r\, r\, \psi_1^*(r) \frac{1}{r^2} \psi_1(r), \tag{B.18}$$

$$\mathcal{I}_3 \equiv \int_0^R \mathrm{d}r\, r\, \psi_1^*(r) \left( -\frac{i}{r} \frac{\partial}{\partial r} \right) \psi_1(r), \tag{B.19}$$

where $R$ is the NW radius.

The parameter $\bar{m}_{m_{\mathrm{F}}}$ in Eq. (B.13), resulting from the matrix elements $\langle \mathrm{i}, m_{\mathrm{F}} | H' | \mathrm{i}, m_{\mathrm{F}} \rangle$, is the harmonic mean of the subband pair between the effective masses $m_{m_{\mathrm{F}},1}$ and $m_{m_{\mathrm{F}},2}$, i.e.,

$$\bar{m}_{m_{\mathrm{F}}} = \frac{2 m_{m_{\mathrm{F}},1} m_{m_{\mathrm{F}},2}}{m_{m_{\mathrm{F}},1} + m_{m_{\mathrm{F}},2}}, \tag{B.20}$$

with

$$m_{m_{\mathrm{F}},1} = \frac{m_0}{\gamma_1 + 2\gamma_{\mathrm{s}}(|c_{m_{\mathrm{F}},1,\mathrm{LH}_\downarrow}|^2 - |c_{m_{\mathrm{F}},1,\mathrm{HH}_\uparrow}|^2)}, \tag{B.21}$$

and

$$m_{m_{\mathrm{F}},2} = \frac{m_0}{\gamma_1 + 2\gamma_{\mathrm{s}}(|c_{m_{\mathrm{F}},2,\mathrm{LH}_\uparrow}|^2 - |c_{m_{\mathrm{F}},2,\mathrm{HH}_\downarrow}|^2)}. \tag{B.22}$$

The parameter

$$\frac{1}{m_{\delta,m_{\mathrm{F}}}} = \frac{m_{m_{\mathrm{F}},2} - m_{m_{\mathrm{F}},1}}{2 m_{m_{\mathrm{F}},1} m_{m_{\mathrm{F}},2}}, \tag{B.23}$$

accounts for the dispersion of this mean and it is typically negligibly small.

Lastly, the effective SOC in Eq. (B.13) is given by

$$
\begin{aligned}
\alpha_{\mathrm{eff},m_{\mathrm{F}}} &= i \langle 1, m_{\mathrm{F}} | H' | 2, m_{\mathrm{F}} \rangle \\
&= \frac{\gamma_{\mathrm{s}}}{m_0} \hbar^2 \sqrt{3}\, \chi_{\mathrm{HH}_\uparrow,\mathrm{LH}_\uparrow}^{m_{\mathrm{F}},1,2} \int_0^R \mathrm{d}r\, r\, \psi_1^*(r) \left[ -ik_r + \frac{1}{2r} - \frac{1}{r}\left( m_{\mathrm{F}} - \frac{1}{2} \right) \right] \psi_1(r) \\
&\quad + \frac{\gamma_{\mathrm{s}}}{m_0} \hbar^2 \sqrt{3}\, \chi_{\mathrm{LH}_\downarrow,\mathrm{HH}_\downarrow}^{m_{\mathrm{F}},1,2} \int_0^R \mathrm{d}r\, r\, \psi_1^*(r) \left[ ik_r - \frac{1}{2r} + \frac{1}{r}\left( m_{\mathrm{F}} + \frac{3}{2} \right) \right] \psi_1(r).
\end{aligned} \tag{B.24}
$$

Notice that, since the Hamiltonian $H_0$ is real and symmetric, we can chose the envelope functions $f_{m_{\mathrm{F}},i,\beta}(r)$ to be real, $\psi_1^* = \psi_1$, and thus $\alpha_{\mathrm{eff}}$ is real too.

These expressions are lengthy and depend on the functional form of the radial wave function $\psi_1(r)$ through the integrals $\mathcal{I}_1$, $\mathcal{I}_2$ and $\mathcal{I}_3$ as well as the envelope functions $f_{m_{\mathrm{F}},i,\beta}$. In principle, they can be found through numerical diagonalization of the Hamiltonian $H_0$ or, as an alternative, we can assume a reasonable functional form for $\psi_1(r)$ and integrate out the radial degrees of freedom to keep the analytical approach. The results of the 8-band k·p calculations in the main text show that the charge density of the NW is almost completely confined in the GaSb shell. Thus, it is reasonable to consider the radial wave function [27],

$$\psi_1(r) = \begin{cases} \sqrt{\frac{2}{wr}} \sin\left[ \frac{\pi}{w}\left( r - R_{\mathrm{av}} + \frac{w}{2} \right) \right], & R - \frac{w}{2} \leq r \leq R, \\ 0, & \text{otherwise}, \end{cases} \tag{B.25}$$

which satisfies hard-wall boundary conditions at $R_{\mathrm{av}} \pm w/2$, being $R_{\mathrm{av}} \equiv R - w/2$ the average effective radius. We note that this ansatz for the radial wavefunction is only valid for a

core-shell NW with a thin $w$ compared to $R$. By plugging Eq. (B.25) in Eqs. (B.17), (B.18) and (B.19), we obtain

$$\mathcal{I}_1 = \left(\frac{\pi}{w}\right)^2, \quad \mathcal{I}_2 = \frac{2\pi}{w^2}\int_0^\pi \frac{\sin(x)^2}{(x-a)^2}, \quad \mathcal{I}_3 = \frac{i}{2}\mathcal{I}_2 - \frac{2\pi i}{w^2}\int_0^\pi \frac{\sin(x)\cos(x)}{(x-a)}, \qquad \text{(B.26)}$$

with $a = (R/w)\pi$. Furthermore, using Eq. (B.25) in Eq. (B.24) we obtain

$$\alpha_{\text{eff}} = \frac{\gamma_s}{m_0}\hbar^2\sqrt{3}\left[\chi_{\text{HH}_\uparrow,\text{LH}_\uparrow}^{m_F,1,2}(1-m_F) + \chi_{\text{LH}_\downarrow,\text{HH}_\downarrow}^{m_F,1,2}(1+m_F)\right]\left(\frac{-2}{w}\right)\int_0^\pi \frac{\sin^2(x)}{(x-a)}\mathrm{d}x. \qquad \text{(B.27)}$$

We take $m_F = \frac{1}{2}$, as we are interested in the fate of the subband pair that can give rise to Majorana zero modes (MZMs). Plugging this equation into Eq. (B.27), we obtain

$$\alpha_{\text{eff},m_F=\frac{1}{2}} = -\frac{\gamma_s}{m_0}\hbar^2\sqrt{3}\left(\chi_\uparrow + 3\chi_\downarrow\right)\frac{1}{w}\int_0^\pi \frac{\sin^2(x)}{(x-a)}\mathrm{d}x, \qquad \text{(B.28)}$$

where

$$\chi_\uparrow = \chi_{\text{HH}_\uparrow,\text{LH}_\uparrow}^{m_F=\frac{1}{2},1,2},$$
$$\chi_\downarrow = \chi_{\text{LH}_\downarrow,\text{HH}_\downarrow}^{m_F=\frac{1}{2},1,2}. \qquad \text{(B.29)}$$

Taking a Taylor expansion of $\frac{1}{x-a}$ up to first order in $|x/a| \ll 1$, the integral in Eq. (B.28) can be approximated by

$$\int_0^\pi \frac{\sin^2(x)}{(x-a)}\mathrm{d}x \sim -\frac{\pi}{2a} - \frac{\pi^2}{4a^2} + \dots, \qquad \text{(B.30)}$$

which gives

$$\alpha_{\text{eff},m_F=\frac{1}{2}} = \frac{\hbar^2\gamma_s}{m_0}\sqrt{3}\chi\left(\frac{1}{R} + \frac{w}{2R^2}\right), \qquad \text{(B.31)}$$

where we define $\chi = (\chi_\uparrow + 3\chi_\downarrow)/2$. This is Eq. (3) of the main text.

In Fig. 3(a) we show the values of $\alpha_{\text{eff},m_F=\frac{1}{2}}$ obtained from Eq. (B.31) with dashed lines, together with the numerical results extracted following the procedure of Appendix A (solid lines). We assume the fixed values $\chi_\uparrow = \chi_\downarrow = 0.3$, corresponding to $c_{1,\text{HH}_\uparrow} = c_{2,\text{HH}_\downarrow} = \sqrt{0.1}$ and $c_{1,\text{LH}_\downarrow} = c_{2,\text{LH}_\uparrow} = \sqrt{0.9}$, for every value of $R$ and $w$, which we extract from our numerical simulations [see Fig. 2(b)]. This gives $\chi = 1.2$.

## C  Effects of strain in the core-shell nanowire

In the previous calculations we have disregarded the effect of the strain that can be present at the core-shell interface of our InP/GaSb NWs. The reason is that strain typically relaxes sharply (in less than 5 atomic layers) at the interface between III-V compound SMs in NWs. Still, we want to quantify how much it could affect the values of SOC we have obtained. Given the similarities between both systems, we can make use of previous analytical results [27] derived for Si/Ge NWs to try to get an estimation in our case.

In the derivations we carried out in Appendix B, we neglected the effect of other subbands that could be close in energy to the first subband pair at finite $k_z$. This was possible for us because we were interested only in the SOC, a property that depends on the vicinity of $k_z = 0$, and our first-order calculations were indeed exact at $k_z = 0$. In Ref. [27], however, a more involved derivation was performed, centered around Ge/Si NWs, including the effect of strain

through the Bir-Pikus Hamiltonian [100]. In their derivation, the authors obtained an effective two-band Hamiltonian that describes the highest-energy hole states of the system through a Schrieffer-Wolff transformation including the effect of the second radial LH subbands as well as the first radial HH subband. Their results are not exact at $k_z = 0$, but they use perturbation theory up to second order (whereas we used first-order perturbation theory). Once this two-band Hamiltonian is projected onto the highest-energy states $|1\rangle$ and $|2\rangle$, and projected onto the radial basis states of Eq. (B.25), they obtain for the $m_F = \frac{1}{2}$ subband pair

$$H_{\text{eff}}^{(s)} \simeq \frac{\hbar^2 k_z^2}{2m_s} \sigma_0 + \alpha_s \sigma_y k_z + \mathcal{O}, \tag{C.1}$$

where the effective mass and SOC in the presence of strain are

$$\frac{1}{m_s} = \frac{1}{m_0} \left[ \gamma_1 + \gamma_s (1 + 3\tilde{\epsilon}_z) - 3\tilde{\gamma} \right], \tag{C.2}$$

$$\alpha_s = \frac{3}{2} \frac{\hbar^2}{m_0 R} \left[ (\gamma_s - \tilde{\gamma}) - (\gamma_1 + \gamma_s)\tilde{\epsilon}_z \right]. \tag{C.3}$$

Above, $\mathcal{O}$ includes all the terms that provide the mean energy and the splitting (see Refs. [26, 27]), which we ignore here for simplicity as they play no role in the present discussion. The renormalized parameters, $\tilde{\gamma}$ and $\tilde{\epsilon}_z$, defined as

$$\tilde{\gamma} = \frac{256}{9\pi^2} \frac{\gamma_s}{10 + \gamma_1(3\epsilon_c + 4\epsilon_r + 2\epsilon_z)/\gamma_s \epsilon_c}, \qquad \tilde{\epsilon}_z = \frac{\epsilon_z}{\epsilon_z + 2\epsilon_r + 2\gamma_s \epsilon_c/\gamma_1}, \tag{C.4}$$

are expressed as a function of the Luttinger parameters $\gamma_1$ and $\gamma_s$, the radial confinement energy $\epsilon_c$, and the strain energies along each direction

$$\epsilon_z = |b| \varepsilon, \quad \epsilon_r = \frac{w}{2} \left( \frac{R + \frac{w}{4}}{\left(R + \frac{w}{2}\right)^2} \right) |b| \varepsilon, \tag{C.5}$$

being $\varepsilon$ the strain coefficient and $b$ the uniaxial deformation potential.

In the absence of strain $\varepsilon = 0$, $\tilde{\epsilon}_z = 0$ and then this Hamiltonian has the same form as the one in Eq. (A.6) and Eq. (B.13). Actually, it provides an estimation for

$$\chi_\uparrow + 3\chi_\downarrow \simeq \sqrt{3} \left( 1 - \frac{256}{9\pi^2} \frac{1}{10 + 3\gamma_1/\gamma_s} \right) = 1.45, \tag{C.6}$$

that applies to the highest-energy mode. Remarkably, this number agrees quantitatively with our numerics, that provided $\chi_\uparrow + 3\chi_\downarrow = 1.2$ (see the discussion at the end of Appendix B). This agreement supports the use of Eq. (A.6) to fit the hole SOC $\alpha$, at least in the absence of strain.

In the presence of strain, it is then reasonable to use the results of Eq. (C.3) to estimate its effect on the SOC of our core-shell NWs. As mentioned above, the GaSb shell is not relaxed close to the interface with the InP insulator, but it is instead strained due to the lattice mismatch between both materials, $\varepsilon = (a_{\text{GaSb}} - a_{\text{InP}})/a_{\text{GaSb}} \simeq 3.93\%$. This provides $\tilde{\epsilon}_z = 0.07$, and a SOC decrease of $\sim 20\%$ very close the interface. This has thus a minor impact in the conclusions of our work.

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
