# Peer review of "InP/GaSb core-shell nanowires: a novel hole-based platform with strong spin-orbit coupling for full-shell hybrid devices"

_SciPost Physics, doi:SciPost Phys. 18, 069 (2025)_

## Round 1 · Referee Report · Anonymous (Referee 1) · 2024-6-17

Report

Vezzosi et al. investigate theoretically using hole bands in core-shell InP/GaSb nanowires incapsulated in a full superconducting shell. Full shell nanowire setups have the advantage of not requiring strong Zeeman energies, rather relying on orbital effects to induce a topological phase. Nonetheless, such setups suffer from the usual issue of distinguishing topological states (MZMs) from trivial. In particular, in the case of full shell nanowires, Caroli–de Gennes–Matricon states can give zero-bias peaks (ZBPs) reminiscent of MZMs. In their paper Vezzosi et al. propose that hole bands in InP/GaAs full shell nanowires could be a better platform than the previously used InAs setups [see Vaitekėnas et al. Science 367, 6485 (2020)], which have attracted both skepticism and controversy. The proposed setup using holes in this paper has the advantage of strong spin-orbit coupling (SOC) than InAs and does not rely on strain or electric fields.

The calculation in this work is solid and a nice example of how core shell nanowires and holes could be a good strategy for achieving strong SOC and, potentially, topological superconductivity. I am minded to eventually accept for SciPost, however I have some concerns since the authors are proposing a very specific platform. In particular, a more in depth discussion of the SC is required before it is possible to really assess the validity of this proposal [see (1) below)]. Nonetheless, if the authors can provide a convincing discussion of the SC then I believe this paper should be published in SciPost and hopefully will encourage experimental efforts.

(1) The authors state “in this work we make a specific and practical proposal“, however throughout — until the final paragraph — I had the question in my of which superconductor was part of this proposal (SC of course being vital for achieving TSC). It is stated at the very end “In any case, the main conclusions of our work should not be affected by the particular SC”. I am not sure I completely agree. Could the authors comment on the following:

1.1) It is stated that metallization effects could be especially important in this setup. I agree with this and think it would be worth spelling out that metallization effects will e.g. likely effectively increase the effective R of the wavefunction and therefore reduce SOC. Similar effects have been seen in before semiconductor and topological-insulator nanowires. I do not think that metallization will negate the benefits of the proposed setup in this paper, but a more in depth discussion of its impact would be useful.
1.2) The TSC region is more than an order of magnitude larger than achieved in the standard Lutchyn-Oreg model, which is the main merit of this proposal. However, Fig. 4 gives the impression that the SC does not matter so much. It would be useful here to briefly mention why it is still beneficial to have a large SC gap (e.g. well localised MZMs etc).
1.3) Although less familiar with attempts to induce superconductivity in GaAs, I worry that many SCs (e.g. Al) will result in a considerable increase in disorder of the GaAs due to the usual mechanisms: diffusion, cross hatched patterns, and spatially dependent interfaces. In assessing the viability of this proposal, it would be very useful for the authors to discuss past literature where such proximity effects have been attempted.

(2) A further concern with the combination of metallization and full shell nanowires is the lack of control over the location of the chemical potential. Other than the larger TSC region (a few meV), it is not really clear to me how this proposal mitigates that issue. Could the authors comment on whether this platform has any benefit (or is further hindered) by the lack of control over chemical potential compared to, say, InAs full shell nanowires?

(3) Another worry is that the limited gating possible in the proposed devices will mainly affect the ends of the nanowire and so will alter the wavefunction primarily in the region at the end. In the proposed setup the conjunction of gating with the likely dependence of metallization on the exact spatial extent wavefunction, seems likely that it could result in many parameters that are rather smooth at the end of the nanowire, which would result in quasi-MBSs. Could the authors comment on if they agree and how to mitigate this? Or do they intend the experiments to be done without any gating?

(4) The extension of the wavefunction around the nanowire core is also reminiscent of topological insulator nanowire. There it was shown that there can be issues inducing superconductivity close to a half-flux quantum, depending on the presence or absence of a vortex in the superconductor and the orbital effect of the states in the TI nanowire [de Juan et al. SciPost Phys. 6, 060 (2019)]. Here it seems there will also be an orbital effect not only in the full SC shell (causing a vortex), but also in the semiconductor. The period of the orbital effects in SC/SM will be different but both related to the radius of the nanowire. I worry that the induced superconductivity might be harmed by the orbital effect of the state in the core-shell nanowire. Could the authors comment on the impact of the orbital effect on induced SC in the nanowire?

(5) Finally, could the authors comment on whether the main experimental signatures of MZMs in the devices they envisage remain those of previous studies: Namely, ZBPs and Coulomb blockade spacing or if there are new signatures possible due to the core-shell nature of the nanowire?

Recommendation

Ask for minor revision

  • validity: top
  • significance: good
  • originality: good
  • clarity: top
  • formatting: perfect
  • grammar: perfect

Author:  Samuel D. Escribano  on 2024-10-22  [id 4889]

(in reply to Report 1 on 2024-06-17)

We thank the Referee very much for his/her positive assessment of our work. As he/she noted, our proposal leverages the strong SOC provided by hole bands in full-shell geometries. Additionally, the core-shell structure significantly suppresses Caroli–de Gennes–Matricon (CdGM) states, reducing the likelihood of misleading them with Majorana bound states (MBSs). This synergy between strong SOC and the suppression of trivial states is key to realizing a robust, gapped topological superconducting phase in our proposed nanodevice. Moreover, we highlight that the Referee thinks that the calculation in our work is solid.

I am minded to eventually accept for SciPost, however I have some concerns since the authors are proposing a very specific platform. In particular, a more in depth discussion of the SC is required before it is possible to really assess the validity of this proposal [see (1) below]. Nonetheless, if the authors can provide a convincing discussion of the SC then I believe this paper should be published in SciPost and hopefully will encourage experimental efforts.

We appreciate that the Referee acknowledges the potential impact of our work in inspiring experimental efforts. We understand his/her concerns regarding the depth of the discussion on the SC part. While the primary challenge of our proposal lies in the SM part of the nanowire, we agree that the SC and the SM/SC interface are critical for achieving a TSC phase. However, we believe that the variety of available superconducting materials offers a broad range of possibilities, and it is largely a matter of experimental exploration, which is difficult to address purely theoretically. Nonetheless, we agree that expanding our discussion on this aspect will strengthen the manuscript. We address these concerns in detail in the following points.

(1) The authors state “in this work we make a specific and practical proposal", however throughout — until the final paragraph — I had the question in my of which superconductor was part of this proposal (SC of course being vital for achieving TSC). It is stated at the very end “In any case, the main conclusions of our work should not be affected by the particular SC”. I am not sure I completely agree. Could the authors comment on the following: 1.1) It is stated that metallization effects could be especially important in this setup. I agree with this and think it would be worth spelling out that metallization effects will e.g. likely effectively increase the effective R of the wavefunction and therefore reduce SOC. Similar effects have been seen in before semiconductor and topological-insulator nanowires. I do not think that metallization will negate the benefits of the proposed setup in this paper, but a more in depth discussion of its impact would be useful. 1.2) The TSC region is more than an order of magnitude larger than achieved in the standard Lutchyn-Oreg model, which is the main merit of this proposal. However, Fig. 4 gives the impression that the SC does not matter so much. It would be useful here to briefly mention why it is still beneficial to have a large SC gap (e.g. well localized MZMs etc). 1.3) Although less familiar with attempts to induce superconductivity in GaAs, I worry that many SCs (e.g. Al) will result in a considerable increase in disorder of the GaAs due to the usual mechanisms: diffusion, cross hatched patterns, and spatially dependent interfaces. In assessing the viability of this proposal, it would be very useful for the authors to discuss past literature where such proximity effects have been attempted.

We apologize for the inaccurate statement, "In any case, the main conclusions of our work should not be affected by the particular SC," as the Referee rightly points out. The choice of SC is indeed critical, and there are specific requirements that it must meet in order to induce a hard superconducting gap, as we detail in that paragraph. However, we believe that there is a broad range of superconducting materials that satisfy these conditions and that can be described using the same Hamiltonian, as happens with electron-based hybrid nanowires. Hence, the particular choice among these SCs is not crucial and it is primarily a matter of experimental preference (or trial and error). To clarify this, we have corrected this sentence to make it clear. Concerning the different points raised above:

1.1) Indeed, metallization effects can be important, as happens also in electron-based hybrid nanowires. In this sense, our proposal does not offer a particular advantage, as far as we can see. The main effects for the topological phase that we expect from metallization are the renormalization of the SOC and the chemical potential of the SM. The Referee mentions another effect, the increase of the effective $R$ of the wavefunction (due to the partial penetration of the SM electron into the SC that gives rise to the proximity effect and, thus, to the metallization). We thank the Referee because we had not thought about this, and it is interesting/relevant for our particular proposal, since the hole SOC depends on R. Let us argue that, since the SC shell in the type of full-shell hybrid nanowires we are investigating in this field are typically very thin (around $5-7$ nm), we don't expect the average wavefunction radius to change very much.

1.2) We agree and we have included this clarification in the revised version.

1.3) The effects of disorder are indeed very important. This is not something particular of our proposal, but in general of Majorana nanowires and related nanostructures. We don't expect our proposal to be better or worse in this respect (although, of course, different materials can demand different SCs). However, as the Referee suggests, we have now included a deeper explanation of the role of disorder due to the SC growth. Firstly, a perfect, clean SC shell is actually detrimental for these hybrid wires, as previous studies have shown (see Ref. [79]). Some degree of disorder, either in the SC itself or at the SM/SC interface, is actually beneficial or even necessary, as it breaks parallel momentum conservation, providing a better hybridization between the SC and the SM and, thus, a good proximity effect. Second, let us mention that nanowires offer a unique advantage compared to other nanostructures as strain is relaxed quickly along the radius due to the large surface-to-volume ratio, reducing strain-disorder related mechanisms. Finally, we remark that most of the mentioned mechanisms are suppressed up to some extent if there is a good matching between the lattice constants of the SC and GaSb. We do mention in our manuscript that "Admittedly, achieving epitaxial growth of Al on GaSb may pose challenges due to the significant lattice mismatch between the two materials. Alternatively, one could explore SCs like Sn [77,78] or Nb [79,80], which have been grown on other III-V compound SMs and posses similar lattice constants as GaSb.'' In any case, if this is not possible, we add "[...] a convenient strategy could be depositing a thin layer of an insulating ternary SM, such as InGaSb, between GaSb and the SC, effectively assimilating the potentially different lattice constants.'' Note that this last strategy would also be convenient to reduce metallization effects, as shown in Phys. Rev. B 107, 184519 (2023). We mention this in the new version of the manuscript.

(2) A further concern with the combination of metallization and full shell nanowires is the lack of control over the location of the chemical potential. Other than the larger TSC region (a few meV), it is not really clear to me how this proposal mitigates that issue. Could the authors comment on whether this platform has any benefit (or is further hindered) by the lack of control over chemical potential compared to, say, InAs full shell nanowires?

Certainly, as the Referee points out, full-shell nanowire devices, including our proposed setup, do not provide direct control over the chemical potential (which is a known drawback of this design, recognized since the Vaitiekenas et al. Science 2020 paper, Ref. [9]). Instead, these devices rely on statistical outcomes. In this sense, our proposal does not improve this known issue of full-shell nanowires. It should be noted, nevertheless, that previous works (Refs. [9, 17]) found that when mode-mixing disorder is taken into account, the topological phase diagram changes so that there is roughly a 50% chance of detecting MBSs across different chemical potentials. Therefore, among multiple devices, one would expect to observe zero-bias peaks (ZBPs) in approximately 50% of the devices, as long as disorder mixes all parallel momenta.

This said, the problem of band alignment and, thus, the control of the chemical potential, is a general problem of hybrid nanowires and something that is still in study in the literature. We would like to point out that there are currently efforts trying to overcome this problem from the material point of view, as can be seen in this recent paper: ``Versatile Method of Engineering the Band Alignment and the Electron Wavefunction Hybridization of Hybrid Quantum Devices", Adv. Mat. 36, 2403176 (2024).

We have added a sentence in the introduction clarifying that our proposal does not solve this known problem of full-shell nanowires, and we have included current strategies citing the above paper (new Ref. [60]).

(3) Another worry is that the limited gating possible in the proposed devices will mainly affect the ends of the nanowire and so will alter the wavefunction primarily in the region at the end. In the proposed setup the conjunction of gating with the likely dependence of metallization on the exact spatial extent wavefunction, seems likely that it could result in many parameters that are rather smooth at the end of the nanowire, which would result in quasi-MBSs. Could the authors comment on if they agree and how to mitigate this? Or do they intend the experiments to be done without any gating?

Certainly, gating at the end of Majorana nanowires (partial or full-shell nanowires) and the smooth variation of parameters (pairing potential, SOC, etc.) when the hybrid region terminates are known to induce quasi-Majorana modes. Some of us explored this issue in much detail in Nat. Rev. Phys. 2, 575 (2020). In short, we don't expect quasi-Majorana physics to differ significantly in our proposal compared to previously studied full-shell nanowires. While quasi-Majorana effects and other forms of disorder along the nanowire's length can harm Majorana physics and warrant further study, they are beyond the scope of this work. Our focus is on proposing an alternative geometry and material for the SM core of the full-shell nanowire, offering key advantages and thus providing a more favorable starting point for exploring Majorana physics. In any case, let us offer some insights on this issue.

Although the problem of quasi-Majorana physics in full-shell nanowires has not yet been analyzed in the literature, it is still possible to conjecture that it presents some advantages with respect to partial-shell nanowires: A) In the bulk of the hybrid nanowire, because the SC shell completely covers the SM, it screens electrostatic fields and electrostatic noise coming from external gates and circuits. Additionally, the strong screening effect helps flatten the electrostatic potential generated by the nanowire’s internal charges. As a result, quasi-Majoranas are less likely to form along the full-shell nanowire, posing a great advantage as compared to partial-shell devices; B) Since the SC completely surrounds the SM, parameter variations at the ends of the hybrid region are expected to occur over shorter length scales compared to partial-shell nanowires. It is well established that quasi-Majoranas tend to form when parameter variations are smooth. The smoother these variations, the more tightly they are pinned near zero-energy as a function of magnetic field or chemical potential, increasing the likelihood of them being mistaken for true MZMs. In contrast, sharp parameter variations, while still capable of producing zero-energy crossings of trivial states (i.e., QD/YSR states), do not result in states that remain pinned to zero energy, making it easier to distinguish them from true MZMs.

Another consideration is whether Majorana physics in full-shell nanowires with a core-shell configuration could be more advantageous (or detrimental) compared to a solid-core one. We don’t expect this to be the case, as quasi-Majorana physics is primarily influenced by parameter variations along the length of the wire, rather than across its cross-section. Therefore, to a first approximation, it is the longitudinal spatial distribution of the wave function, and not transverse, the one that is most relevant.

A further discussion is whether metallization might more easily degrade the properties of the SM in a core-shell geometry compared to a full-shell one. However, this issue affects the entire wire and has been addressed in previous discussions. Generally, metallization effects should be minimized to preserve the essential qualities of the SM required for topological physics, such as strong SOC. This issue lies beyond the scope of the current study, as it would depend on the specific SM/SC heterojunction and the potential inclusion of an insulating layer between them, if necessary.

We have added a sentence in the introduction to clarify this important point.

(4) The extension of the wavefunction around the nanowire core is also reminiscent of topological insulator nanowire. There it was shown that there can be issues inducing superconductivity close to a half-flux quantum, depending on the presence or absence of a vortex in the superconductor and the orbital effect of the states in the TI nanowire [de Juan et al. SciPost Phys. 6, 060 (2019)]. Here it seems there will also be an orbital effect not only in the full SC shell (causing a vortex), but also in the semiconductor. The period of the orbital effects in SC/SM will be different but both related to the radius of the nanowire. I worry that the induced superconductivity might be harmed by the orbital effect of the state in the core-shell nanowire. Could the authors comment on the impact of the orbital effect on induced SC in the nanowire?

For a full-shell hybrid nanowire, accounting for orbital effects in the SM is essential for Majorana physics, a topic extensively discussed in the literature (see Refs. [9, 17, 24]). Let us clarify this further, particularly in comparison to the work of SciPost Phys. 6, 060 (2019).

As the Referee correctly notes, a vortex (or more precisely, an odd number of vortices) in the SC and appropriate orbital conditions in the nanowire are required to generate MBSs. The magnetic flux $\Phi$ through the cylindrical SC shell creates in general a quantization of the fluxoid $\Phi'$ in units of the SC flux quantum $\Phi_0=h/2e$, i.e., $\Phi'=n \Phi_0$, being $n$ the winding of the superconducting phase around the axis (and thus an integer). We are interested in a thin SC shell to preserve its superconductivity in the presence of a magnetic field. In this case, the so-called Little-Parks (LP) effect appears. Notice that the flux is not quantized through the SC tube (only the fluxoid), and therefore we have a window of magnetic fluxes within each LP lobe labeled by $n$.

This quantized fluxoid is induced in the nanowire through the proximity effect, and the total angular-momentum quantum number, $m_J=m_L+m_S+\frac{n}{2}$ (where $m_L$ and $m_S$ are the orbital and spin quantum numbers, respectively), must be conserved for a cylindrical full-shell nanowire. A MBS must have $m_J=0$ so that it can be its own self-charge conjugate. This can only occur if $n$ is odd, as only in those cases $m_J$ can be an integer. Therefore, MBSs can appear only in the odd number LP lobes. The average radius of the SC, among other parameters, determines the magnetic field windows of each lobe.

In addition, in order to have Majorana zero-energy modes, the lowest $m_J=0$ subband must cross zero energy and undergo a topological phase transition (inside of an odd lobe). One might consider to tune the energy position of the $m_J$ levels through a gate potential or a Zeeman field. However, in full-shell geometries, the former is not feasible, and the latter is typically negligible in the first LP lobes, specially for large $R$, which are the most accessible ones. Nonetheless, as demonstrated in Refs. [9, 17], the orbital effect of the magnetic field creates an effective Zeeman field that strongly tunes the different $m_J$ subbands within each $n$-lobe. In full-shell nanowires, it is the orbital effect in the SM, in combination with the induced fluxoid, what enables the conditions for a topological phase.

If the superconducting shell is thin compared to the London penetration length, the magnetic flux inside the SM is not quantized, as we mentioned above, and the orbital-induced effective Zeeman field evolves continuously with the applied magnetic field within each lobe. It is possible to demonstrate that, in the topological phase (i.e., for appropriate nanowire parameters), the flux extension of the Majorana zero-energy mode within the odd lobes is determined by the average radius of the SM wavefunction. The greater the difference between the average radii of the SM and SC, the larger the Majorana zero mode extension within the 1st lobe (and subsequent odd lobes), though at the expense of reduced topological protection (see Ref. [17] for a detailed analysis). In other words, a mismatch of the period with flux between the SC shell features and the SM subgap features is not only possible, but it leads to wider Majorana zero-energy mode flux intervals within odd lobes.

In contrast, in the system discussed by SciPost Phys. 6, 060 (2019), the superconducting shell is thick, leading to a quantized magnetic flux within the SM and a corresponding quantization of the orbital-induced Zeeman field. Consequently, we believe that the difference is that the conditions for a topological superconducting phase are stricter, as one of the tunable parameters is effectively constrained.

Despite the importance of orbital effects for Majorana physics, we do not analyze them in this work for two reasons: (i) Although the Majorana physics is the motivation of our proposal, we don't explicitly analyze it in this work; and (ii) they warrant a separate study, given the complex spin and orbital structure of the hole bands. However, in systems like ours, where confinement strongly splits the LH and HH bands, we expect the orbital effects to behave similarly to those in conventional full-shell nanowires, contributing to the emergence of the topological superconducting phase.

We have added a brief explanation of this point in the revised manuscript and cited SciPost Phys. 6, 060 (2019) (new Ref. [72]).

(5) Finally, could the authors comment on whether the main experimental signatures of MZMs in the devices they envisage remain those of previous studies: Namely, ZBPs and Coulomb blockade spacing or if there are new signatures possible due to the core-shell nature of the nanowire?

Although a good point, we do not anticipate new experimental signatures specific to the core-shell structure of the nanowire. All the well-established methods from previous studies, such as detecting zero-bias peaks (ZBPs) and Coulomb blockade spacing in conductance experiments, remain applicable to our nanodevices. That said, we believe our design may reduce the likelihood of false positives compared to other setups, as explained in the main text. We have made this aspect clear in the introduction of the new manuscript.

---

## Round 1 · Referee Report · Anonymous (Referee 2) · 2024-7-2

Report

In this work, the authors theoretically study InP/GaSb core-shell hole-band nanowires as a candidate material for the semiconductor part of full-shell hybrid superconductor-semiconductor (SC-SM) Majorana nanowires. The band structure of the bare InP/GaSb core-shell nanowire close to the topmost valence band is obtained from an 8-band Kane model through a self-consistent solution of the Poisson equation. The main finding is that the GaSb holes exhibit a strong intrinsic spin-orbit interaction that does not rely on electric fields. The authors also calculate the size of the ‘topological chemical-potential window’, which is the energy splitting between the two highest-energy hole subbands with total angular momentum quantum number |mF|=1/2. Based on their results, the authors argue that InP/GaSb core-shell hole-band nanowires are a promising candidate platform for full-shell Majorana physics.

The paper is interesting and well-written, and the presented band structure calculations for the bare InP/GaSb core-shell nanowire appear to be technically sound. However, I have a couple of concerns/questions related to the Majorana part of the work that should be addressed before I can recommend publication in SciPost Physics:

1) I think that the current title—and to some extent also the abstract—of the manuscript do not reflect its content very accurately. The reasons for this are the following:

  • The paper does not discuss the actual Majorana physics in the SC-SM full-shell hybrid nanowire in any detail, but only the normal-state band structure of the bare InP/GaSb core-shell nanowire, which is why I think it is not appropriate to call the paper a “proposal for Majorana modes”. In particular, the authors neither write down a superconducting pairing term nor a time-reversal symmetry breaking term (magnetic flux), both of which are necessary for the emergence of Majorana zero modes in full-shell SC-SM hybrid nanowires. Instead, when it comes to the emergence of Majorana modes, the authors just cite a couple of previous works without any additional explanations. If the paper should be called a “proposal for Majorana modes”, I think it would be important to (i) explicitly specify the full Hamiltonian of the SC-SM hybrid, and (ii) substantiate the Majorana part of the work by additional calculations. Alternatively, the title of the paper could be changed.

  • I also think the use of the word ‘practical’ in the title is slightly misleading. It appears that InP/GaSb core-shell nanowires have not yet been fabricated, and even if they can be fabricated in the future, it is not clear whether their quality will be high enough to overcome the usual disorder problems that plague more conventional (e.g. InAs-based) Majorana nanowire platforms. It is also not clear which superconductor can be used to induce a (hard) superconducting gap for the holes in the GaSb shell, nor is it known what the size of the induced gap would be. Taking all of these points together, the setup proposed here should in my opinion not be called ‘practical’ as none of the necessary ingredients are currently available in the lab.

  • The abstract gave me the impression that CdGM states (and how their presence can potentially be reduced in a core-shell geometry) would be discussed in this paper, but this is not the case. Instead, CdGM states only enter the paper through references to previous works (mainly Ref. [17]). Maybe the authors can slightly rephrase the abstract in order to avoid this potential misunderstanding.

2) One important assumption made by the authors is that “the Fermi level is placed close to the VB edge of GaSb as reported in Ref. [28].” However, is there any reason to assume that the Fermi level remains close to the valence band edge in the presence of a superconducting shell? It does not seem unlikely that the Fermi level is shifted substantially in the presence of an SC shell (especially in the limit of strong SC-SM coupling, which is often the experimentally relevant limit). If this is the case, it is not clear whether the chemical potential can ever fall into the ‘topological chemical-potential window’ calculated by the authors (see Fig. 4 in the manuscript) since the Fermi level cannot be adjusted by gating in a full-shell geometry. It would be good if the authors could comment on this potential problem.

3) The radii of the wires considered in this work (R=15-40 nm) are relatively small compared to the radii considered in previous works on InAs electron nanowires (e.g. R~65 nm in Ref. [9]). For such small nanowire radii, the magnetic field needed to thread one flux quantum through the SC shell can become sizable. Can the authors give an estimate for the approximate field strength that is needed to tune their system deep into the topological phase? I wonder how much the band structure shown in Fig. 2 will be modified by the magnetic field in the SM if both Zeeman and orbital effects are taken into account. Especially orbital effects might be important in a core-shell geometry, see also point (4) of the first Referee.

4) I think there may be a typo on page 7: "As in the case of Ref. [20], we find that α increases with R." Shouldn't it be "[...] decreases with R"?

Recommendation

Ask for minor revision

  • validity: high
  • significance: good
  • originality: good
  • clarity: high
  • formatting: perfect
  • grammar: perfect

Author:  Samuel D. Escribano  on 2024-10-22  [id 4888]

(in reply to Report 2 on 2024-07-02)

We thank the Referee very much for pointing out the importance of our work. As he/she mentions, our work makes an effort to try to capture all the complexity of the band structure of the semiconductor, offering reliable outcomes. In addition, we would like to highlight that the Referee thinks that the calculations appear to be technically sound.

However, I have a couple of concerns/questions related to the Majorana part of the work that should be addressed before I can recommend publication in SciPost Physics:

1) I think that the current title—and to some extent also the abstract—of the manuscript do not reflect its content very accurately. The reasons for this are the following:

- The paper does not discuss the actual Majorana physics in the SC-SM full-shell hybrid nanowire in any detail, but only the normal-state band structure of the bare InP/GaSb core-shell nanowire, which is why I think it is not appropriate to call the paper a “proposal for Majorana modes”. In particular, the authors neither write down a superconducting pairing term nor a time-reversal symmetry breaking term (magnetic flux), both of which are necessary for the emergence of Majorana zero modes in full-shell SC-SM hybrid nanowires. Instead, when it comes to the emergence of Majorana modes, the authors just cite a couple of previous works without any additional explanations. If the paper should be called a “proposal for Majorana modes”, I think it would be important to (i) explicitly specify the full Hamiltonian of the SC-SM hybrid, and (ii) substantiate the Majorana part of the work by additional calculations. Alternatively, the title of the paper could be changed.

- I also think the use of the word ‘practical’ in the title is slightly misleading. It appears that InP/GaSb core-shell nanowires have not yet been fabricated, and even if they can be fabricated in the future, it is not clear whether their quality will be high enough to overcome the usual disorder problems that plague more conventional (e.g. InAs-based) Majorana nanowire platforms. It is also not clear which superconductor can be used to induce a (hard) superconducting gap for the holes in the GaSb shell, nor is it known what the size of the induced gap would be. Taking all of these points together, the setup proposed here should in my opinion not be called ‘practical’ as none of the necessary ingredients are currently available in the lab.

Following the Referee suggestion, in order to avoid any confusion, we have decided to amend our original title to "InP/GaSb core-shell nanowires: a novel hole-based platform with strong spin-orbit coupling for full-shell hybrid devices". Although our former title was more specific (and more ambitious) regarding the main motivation and potential application of our work, we agree that, since topological superconductivity (TSC) is not actually studied in the manuscript, the title could be misleading. The new title keeps it more general, addressing that we study a novel SM heterostructure that could be used to fabricate full-shell hybrids with hole bands.

- The abstract gave me the impression that CdGM states (and how their presence can potentially be reduced in a core-shell geometry) would be discussed in this paper, but this is not the case. Instead, CdGM states only enter the paper through references to previous works (mainly Ref. [17]). Maybe the authors can slightly rephrase the abstract in order to avoid this potential misunderstanding.

We understand the confusion of the Referee, and now we have rephrased the abstract to make clear that we do not analyze CdGM states.

2) One important assumption made by the authors is that “the Fermi level is placed close to the VB edge of GaSb as reported in Ref. [28].” However, is there any reason to assume that the Fermi level remains close to the valence band edge in the presence of a superconducting shell? It does not seem unlikely that the Fermi level is shifted substantially in the presence of an SC shell (especially in the limit of strong SC-SM coupling, which is often the experimentally relevant limit). If this is the case, it is not clear whether the chemical potential can ever fall into the ‘topological chemical-potential window’ calculated by the authors (see Fig. 4 in the manuscript) since the Fermi level cannot be adjusted by gating in a full-shell geometry. It would be good if the authors could comment on this potential problem.

The SC shell will indeed change the Fermi level of the heterostructure because of the work function difference and the band offset between both materials. How much depends on numerous experimental factors beyond the chemical properties of the materials, such as system dimensionality, interface quality, growth process, and fabrication method. Therefore, this issue must be addressed experimentally. On top of this, there can be metallization problems as Referee 1 mentioned in question (2). So, in general, in full-shell hybrid nanowires, the modus operandi nowadays is to grow many wires and measure them, with the expectation that some of them will be in the correct chemical potential window. In fact, there is the equivalent problem with electron-based nanowires, the Fermi level needs to be close to the conduction band edge in that case, see for instance the analysis of Ref. [20]. Since this is in general a problem of hybrid wires, we would like to point out that there are currently efforts trying to overcome this problem from the material point of view, as can be seen in this recent paper: "Versatile Method of Engineering the Band Alignment and the Electron Wavefunction Hybridization of Hybrid Quantum Devices", Adv. Mat. 36, 2403176 (2024). Please, see also our answer to Referee 1 (question 2) for more details.

This said, in the particular case of core-shell nanowires, as the ones that we study here, as we mention in the text: Partial control over the wire’s overall doping could be achieved through the incorporation of chemical impurities within the InP core during the growth process [26, 58, 59]. Given that the InP core primarily acts as an insulator in this context, it is reasonable to assume that these dopants would exert a minimal influence on the electronic properties of the wire, aside from their impact on the Fermi level position.

In the new version of the manuscript, we now discuss further this problem, cite the above reference, and write explicitly that the effect of the SC outer shell on the Fermi level position could be corrected through the chemical impurities mechanism.

3) The radii of the wires considered in this work ($R=15$-$40$ nm) are relatively small compared to the radii considered in previous works on InAs electron nanowires (e.g. $R\simeq 65$ nm in Ref. [9]). For such small nanowire radii, the magnetic field needed to thread one flux quantum through the SC shell can become sizable. Can the authors give an estimate for the approximate field strength that is needed to tune their system deep into the topological phase? I wonder how much the band structure shown in Fig. 2 will be modified by the magnetic field in the SM if both Zeeman and orbital effects are taken into account. Especially orbital effects might be important in a core-shell geometry, see also point (4) of the first Referee.

This is a good point. The magnetic field required to drive the system into the TSC phase depends on the choice of the SC material, as the Little-Park effect is influenced by the SC’s critical temperature and diffusive coherence length. For instance, with Al as the SC shell, we estimate that a magnetic field of approximately $0.2$ T would be sufficient to reach the center of the first lobe for a nanowire with a radius of $40$ nm, whereas a field of $1.7$ T would be required for a nanowire with a radius of $15$ nm. The latter might be challenging, as high magnetic fields tend to destroy the superconductivity of the parent SC. Therefore, a thicker nanowire is advantageous in this regard. So, there is a trade-off between having thinner nanowires to achieve larger SOCs, and thicker ones to require smaller applied magnetic fields. We have added this important information in the discussion we make in Section 4.

As for the orbital effects in the nanowire, we do not analyze them in this work since they warrant a separate study due to the complex spin and orbital structure of the hole bands. Nevertheless, for a system like ours, where confinement induces a strong splitting between LH and HH bands, we expect the orbital effects to behave similarly to those in conventional full-shell nanowires, driving the emergence of the TSC phase. We note that, as opposed to partial-shell nanowires, in full-shell nanowires the orbital effects are necessary for the topological phase. Please, see the answer to Referee 1, question (4), for a more in-depth discussion. On the other hand, the Zeeman effect of the magnetic field is not necessary for the topological phase in full-shell geometries. But, if present, it contributes positively to it, meaning that it widens the topological regions of the topological phase diagram (see Ref. [17]). It is typically disregarded because it is small in the first LP lobe, albeit it should be included.

4) I think there may be a typo on page 7: "As in the case of Ref. [20], we find that $\alpha$ increases with $R$." Shouldn't it be "[...] decreases with $R$"?

Thanks for spotting this typo. The reviewer is right: in both works, ours and in Ref. [20], $\alpha$ decreases as $R$ increases. We have fixed it in our new version of the manuscript.

---

## Round 1 · Referee Report · Anonymous (Referee 3) · 2024-7-18

Strengths

  • alternative proposal for a full-shell nanowire structure with potentially larger spin-orbit and hence larer topological gap

Weaknesses

  • unclear description of the spin-orbit effect in the paper
  • conclusions on the origin of the spin-orbit effect are doubtful

Report

My apologies for the late report.

This paper deals with an alternative proposal for realizing a full-shell nanowire to create Majorana zero modes. The authors in particular argue that using holes in the valence band gives rise to a larger spin-orbit coupling and hence a larger topological gap.

The other referees have already mentioned several aspects related to using the core-shell nanowire in a full-shell structure for Majorana zero modes, and I agree with their assessment there. In addition, I find the discussion of the main result, the large spin-orbit coupling at times confusing and I am not convinced by some of the conclusions the authors draw in this paper. In particular:

  1. Readability of the manuscript: The authors do not properly define what they mean with spin-orbit interaction. Since the system has inversion symmetry and time-reversal symmetry, there can be no k-dependent spin splitting of the bands, and indeed this also does not appear in the numerics. However, this is what a reader would expect when hearing the word Rashba SOI in isolation. The authors comment a bit on this on page 7 on saying the Rashba field is radial, but this is rather vague. To make things worse, when the authors write down an effective Hamiltonian, e.g. Eqs. (1) and (9), the usual Rashba term $\alpha k_z \sigma_y$ arising from an electric field appears, which would lead to a k-dependent spin-splitting! However, this is not what e.g. Ref. [20] uses! There the Hamiltonian is $\alpha \frac{\vec{r}}{r} \cdot (\vec{k} \times \vec{\sigma})$. This is deeply confusing. Is this just sloppy notation? Is the derivation of the effective model wrong?
  2. Interpretation of origin of spin-orbit coupling: The authors write "These results lead us to conclude that the origin of the SOC is an inherent property of GaSb, rooted in the symmetry of the crystal structure of this tetravalent SM." I don't find this convincing at all. The model used by the authors is a generic k.p model that applies to a wide range of III-V semiconductors, i.e. their results do not seem specific to GaSb. Furthermore, for the effective model the authors find the effect even in spherical approximation, meaning that the effect does not depend on the crystal structure. To me it seems more likely, that the band edge discontinuity at the core plays a similar role to the band bending in the previous studies (albeit now a discrete jump). The size of the effect can de bigger, as this for holes (similar to what is called direct Rashba SOI in Ref. [50]). To this end it would be interesting to see if the effect also appears as the size of the core shrinks to 0 (no core). This would mean to extend Fig. 3 to zero core radius (or w=R). Either there is a maximum of the spin-orbit and goes to zero as the core vanishes (then the origin is the discontinuity at the core) or there is a finite strength even for no core, and then this would mean the effect is is due to the confinement of the nanowire itself.

Since these questions affect the very core of the paper, I cannot formulate any recommendation to publish until they are resolved.

Recommendation

Ask for major revision

  • validity: poor
  • significance: ok
  • originality: good
  • clarity: poor
  • formatting: good
  • grammar: good

Author:  Samuel D. Escribano  on 2024-10-22  [id 4890]

(in reply to Report 3 on 2024-07-18)

We appreciate the time the Referee has taken to consider our work and we thank him/her for his/her feedback. We now recognize that the origin of the SOI in our system was not sufficiently well explained, which we believe lead to the confusion of the Referee. We provide below detailed answers to his/her questions, and we have included pertinent explanations in the revised version of the manuscript. We hope that with this we can convince the Referee of the validity of our methods and results.

1. Readability of the manuscript: The authors do not properly define what they mean with spin-orbit interaction. Since the system has inversion symmetry and time-reversal symmetry, there can be no $k$-dependent spin splitting of the bands, and indeed this also does not appear in the numerics. However, this is what a reader would expect when hearing the word Rashba SOI in isolation. The authors comment a bit on this on page 7 on saying the Rashba field is radial, but this is rather vague. To make things worse, when the authors write down an effective Hamiltonian, e.g. Eqs. (1) and (9), the usual Rashba term $\alpha k_z \sigma_y$ arising from an electric field appears, which would lead to a $k$-dependent spin-splitting! However, this is not what e.g. Ref. [20] uses! There the Hamiltonian is $\alpha \frac{\vec{r}}{r}\cdot (\vec{k}\times\vec{\sigma})$. This is deeply confusing. Is this just sloppy notation? Is the derivation of the effective model wrong?

Indeed, we did not make the point clear, which has led to some misunderstanding. Let us clarify all the points raised above by the Referee. First of all, our definition of the SOI is the conventional one and the one that the Referee mentions: a $k$-dependent spin-splitting of the bands. This arises in our system because there is indeed a spatial symmetry that is broken: confinement breaks the symmetry of the bulk crystal (see more details in the next question). This kind of SOI, which does not arise because of a structural or bulk inversion asymmetry, has been widely studied before [see, for instance, Ref. [52] for a work with similar system to ours, particularly Eqs. (12,13)].

Secondly, we do clearly observe this spin-splitting in our numerical calculations, as shown in the band structure in Fig. 2(b) and the extracted values of the SOC in Fig. 3. The Referee might be misinterpreting the behavior of the system due to the cylindrical symmetry of our nanostructure. In such systems, subband pairs affected by SOC are not degenerate at $k_z=0$ because the angular momentum opens a small gap. However, SOC can still be identified by the differing slopes of subbands within the same pair as a function of $k_z$. For a detailed explanation of this behavior, we refer the Referee to the Supplemental Material of Ref. [20], from where we extract the method to infer the SOC from comparable band structures.

Third, as indicated in our manuscript, the Hamiltonian describing our bands does have the form of the Rashba type although it is not originated from a Rashba field. This is because the coefficient $\alpha$ does not originate from an electric field, but rather from the degree of hybridization between the HH and LH bands. This Hamiltonian is not ad hoc, we derive it from the full Luttinger-Kohn Hamiltonian that we project onto the lowest-energy subbands (see Appendix B). We prove that this analytical effective Hamiltonian reproduces the bands that we obtain numerically, and therefore provides a reliable description of the bands. This procedure clearly shows that the SOI arises due to the intrinsic SOI of GaSb and the nanostructure radial confinement, leading to a $k$-dependent spin-splitting, akin to a conventional linear SOI.

As a matter of fact, albeit they are not the same because they have different origins, the Hamiltonian of our effective SOI and the one used in Ref. [20] turn out to have the same form, although notice that they are expressed in different bases. As explained in our Appendix B, our Hamiltonian is written in a rotated basis independent of the angle. If we take the SOI from Ref. [20], $H_{\rm SOI}=\vec{\alpha}\cdot(\vec{k}\times\vec{\sigma})=\alpha_r(k_\phi \sigma_z-k_z\sigma_\varphi)$, and rotate it with $U=e^{i\left(m_{\rm J}-\frac{1}{2}\sigma_z\right)\varphi}$, we obtain $ U^\dagger H_{\rm SOI} U= \frac{\alpha_r}{r}\left(m_{\rm J}-\frac{1}{2}\sigma_z\right)-\alpha_r k_z\sigma_y$. The second term corresponds to our expression for the effective SOI. Also, notice that the first term reveals the broken degeneracy at $k_z=0$ that we mentioned. This clearly indicates that our SOC is radial, too, which makes sense as confinement breaks the symmetry along the radial direction.

We have added several sentences in Section 3.3 and 3.4 to make these points clear.

2. Interpretation of origin of spin-orbit coupling: The authors write "These results lead us to conclude that the origin of the SOC is an inherent property of GaSb, rooted in the symmetry of the crystal structure of this tetravalent SM." I don't find this convincing at all. The model used by the authors is a generic k$\cdot$p model that applies to a wide range of III-V semiconductors, i.e. their results do not seem specific to GaSb. Furthermore, for the effective model the authors find the effect even in spherical approximation, meaning that the effect does not depend on the crystal structure. To me it seems more likely, that the band edge discontinuity at the core plays a similar role to the band bending in the previous studies (albeit now a discrete jump). The size of the effect can de bigger, as this for holes (similar to what is called direct Rashba SOI in Ref. [51]). To this end it would be interesting to see if the effect also appears as the size of the core shrinks to 0 (no core). This would mean to extend Fig. 3 to zero core radius (or w=R). Either there is a maximum of the spin-orbit and goes to zero as the core vanishes (then the origin is the discontinuity at the core) or there is a finite strength even for no core, and then this would mean the effect is is due to the confinement of the nanowire itself.

Again, we believe that we simply have not expressed ourselves clearly in the previous version, and this has caused the misunderstandings expressed above. We are sorry for that. Let us clarify several aspects.

Indeed, our Hamiltonian is general and applies to all III-V SM compound heterostructures. Actually, all of them exhibit this type of SOI in nanostructures. However, our quantitative results are specific to the GaSb/InP heterostructure that we analyze here, as we use the particular parameters of these materials. In Section 2 of our manuscript, we provide a discussion about why we expect other III-V SM compound heterostructures are less suitable or exhibit poorer performance. For example, some would provide a too small SOI, in others the Fermi level wouldn't lie at the valence bands, or the core and shell bands would be mixed, etc.

Apart from this, the spherical approximation applies to the envelope function k, not the crystal momentum p. In a k$\cdot$p model, these two quantities warrant different treatment. Thus, even though the mesoscopic system has cylindrical symmetry, the underlying crystal does not possess the same symmetry.

Regarding the origin of the SOC, we want to clarify that it cannot be attributed to band discontinuities, which in any case are fundamentally different to a band-bending. We devote part of our work to explore this situation, and the results are illustrated in Fig. 3(d) of our manuscript. Our analysis concludes that the SOC is intrinsically related to the GaSb material and its finite size effects, rather than to band/parameter discontinuities. In fact, our findings show that even if we completely remove the core of the nanowire (not shown because it wouldn't be a core-shell structure), the effective SOC for a subband does not vanish. This observation reinforces that the SOC is a characteristic inherent to radial confinement in the GaSb material and its specific properties, rather than being solely a result of the core-shell interface.

To better understand the origin of the SOC it is enlightening to look into the analytical derivation we perform in Appendix B. Notice that there is an intrinsic SOI in the bulk of III-V SM compounds arising from the orbital SOI of the crystal atoms. This actually manifests by splitting the valence bands into HH and LH bands. However, this intrinsic SOI in the bulk does not cause spin-splitting within the HH or LH bands themselves (i.e., no splitting between spin-up and spin-down states within a single band). Instead, the $k$-dependent spin-splitting separates bands with different total angular momentum values, i.e., HH with $m_{\rm S}=\frac{3}{2}$ and LH with $m_{\rm S}=\frac{1}{2}$ are split. In a nanostructure, confinement breaks the translational symmetry of the bulk crystal, leading to quantization of the momentum in the confined directions. This quantization can mix the HH and LH bands, which are now no longer strictly distinguishable by their angular momentum projections $m_{\rm S}$. As a result, the subbands, which are hybridized mixtures of HH and LH, may exhibit $k$-dependent spin-splitting among themselves.

We find analytically the expression of the SOI for the lowest-energy subbands in a nanowire with cylindrical symmetry. We find it is indeed proportional to the degree of hybridization between HH and LH (with same spin), and inversely proportional to the confined dimension (the radius in our case). Importantly, we are not the first ones reporting this SOC, but it is a known effect. For instance, in Ref. [52] the authors analyze exactly the same problem as us and, although their perturbative method is slightly different, they arrive to a similar expression for the SOC [see Eq. (13) there, or our discussion in Appendix C].

This SOC does not correspond to a conventional Rashba SOC, which is proportional to an electric field and arises from a structural inversion asymmetry; neither a Dresselhaus SOC, which arises from bulk inversion asymmetry in the crystal lattice itself. Instead, the $k$-dependent spin-splitting occurs due to the combination of quantum confinement and the intrinsic SOI of the material. Precisely because this SOI is "intrinsic" and only depends on the dimensionality of the system, this nanodevice is so advantageous for full-shell physics.

We now include this more detail description of the origin of the SOI in our manuscript, both, in Section 3.3 and 3.4 (where the origin of the SOC is discussed) and in Section 4 (Conclusions).

---

## Round 2 · Referee Report · Anonymous (Referee 1) · 2024-11-11

Report

I feel that the authors have largely addressed my concerns. I think the paper is of interest for future studies of full shell nanowires and potential MBSs in these systems, however for the paper to be of significant value then an analysis of the superconducting properties would have strengthened it considerably (and was what the original paper promised). Without that discussion I think the current paper just about meet SciPost's requirements.

Recommendation

Publish (meets expectations and criteria for this Journal)

---

## Round 2 · Referee Report · Anonymous (Referee 2) · 2024-11-20

Report

The authors have addressed all comments in my previous report satisfactorily. I think that the revised version of the manuscript now meets the acceptance criteria of SciPost Physics.

Recommendation

Publish (meets expectations and criteria for this Journal)

---

## Round 2 · Referee Report · Anonymous (Referee 3) · 2024-11-25

Report

I am very grateful to the authors for the extensive reply as well as the revised manuscript! These have helped to clear up some of my previous confusion - I was under the impression that the interaction described by alpha was within one subband, not between subbands.

I understand now the authors' results, and I think they are interesting. My main question however is the following: The main result, a coupling between the two lowest m_F=1/2 subbands as in Eq. (1) has already been shown in Eq. (3) in Ref. [50] (Kloeffel et al., PRB 84, 195314 (2011)): the term with the $C$ prefactor is identical to what the authors find.

This thus raises a major question: The paper positions itself mainly as showing strong spin-orbit in this proposed system. A very similar result (same order of magnitude) has previously been demonstrated for Ge, a platform that already exists. Is this sufficient to claim "Open a new pathway in an existing or a new research direction, with clear potential for multi-pronged follow-up work"?

Currently, Ref. [50] is somewhat dismissed in the beginning of the paper as "However, these favorable properties stem from the strain introduced at the Ge/Si heterostructure interface [50–52], which imposes specific design constraints." The term in Ref. [50] that corresponds to the author's results is without strain, however. Hence, I believe the relationship to this (and possibly other papers) is not properly discussed in this paper. This also hinders me in making an assessment where this paper should be published.

If the authors can make a compelling argument for why their proposal is fundamentally better than the previous literature, then I would suggest publication in SciPost Physics. If not, then I would propose to publish in SciPost Physics Core. In both cases, I would like to ask the authors to consider my other points listed below.

Other points with regards to this manuscript: - I misunderstood the paper when I read it first. There is a reason for this: the authors use symbols as for the conduction band case ($\alpha$) and initially do not define what they mean with "spin-orbit" (this has only slightly improved in the revised version). In fact, Ref. [50] has the same term but does not call it spin-orbit interaction (Ref. [50] later derives a Rashba type SOI). I would suggest to the authors to (i) define what they call spin-orbit already in the beginning of Sec. 3.2, and not just in the appendix (and even with the appendix, I had to read all the previous papers to understand what was going on - it would be great to make this more self-contained), (ii) give an argument why they call this spin-orbit interaction. - I find App. B a stronger argument for claiming that the effect is due to the Hamiltonian and not electric field than all the arguments in Sec. 3.2. Basically, Sec. 3.2 shows through numerics what does not affect the strength of the effect. App. B shows you get the effect from the Luttinger-Kohn Hamiltonian and confinement alone. To me, it would make more sense to state this fact first, and later show that the more detailed numerics only gives small corrections. - Why do the authors consider an insulating core at all? It is clear from the results that the effect they want to show also will appear without an insulating core (as is the case in Ref. [50] for example)

Recommendation

Ask for major revision

  • validity: good
  • significance: ok
  • originality: good
  • clarity: low
  • formatting: good
  • grammar: excellent

Author:  Samuel D. Escribano  on 2024-12-18  [id 5053]

(in reply to Report 3 on 2024-11-25)

I am very grateful to the authors for the extensive reply as well as the revised manuscript! These have helped to clear up some of my previous confusion - I was under the impression that the interaction described by alpha was within one subband, not between subbands. I understand now the authors' results, and I think they are interesting.

We sincerely appreciate the Referee's positive feedback and we are glad that our revised manuscript has helped clarify the nature of the interaction described by $\alpha$, thereby enhancing our manuscript readability. We are also pleased to hear that the Referee finds our results interesting. This aligns well with the feedback from the other Referees, who similarly highlighted the novelty and significance of our work.

My main question however is the following: The main result, a coupling between the two lowest $m_F=1/2$ subbands as in Eq. (1) has already been shown in Eq. (3) in Ref. [50] (Kloeffel et al., PRB 84, 195314 (2011)): the term with the C prefactor is identical to what the authors find.

This thus raises a major question: The paper positions itself mainly as showing strong spin-orbit in this proposed system. A very similar result (same order of magnitude) has previously been demonstrated for Ge, a platform that already exists. Is this sufficient to claim "Open a new pathway in an existing or a new research direction, with clear potential for multi-pronged follow-up work"?

Currently, Ref. [50] is somewhat dismissed in the beginning of the paper as "However, these favorable properties stem from the strain introduced at the Ge/Si heterostructure interface [50–52], which imposes specific design constraints." The term in Ref. [50] that corresponds to the author's results is without strain, however. Hence, I believe the relationship to this (and possibly other papers) is not properly discussed in this paper. This also hinders me in making an assessment where this paper should be published.

If the authors can make a compelling argument for why their proposal is fundamentally better than the previous literature, then I would suggest publication in SciPost Physics. If not, then I would propose to publish in SciPost Physics Core.

We thank the Referee for their insightful comments, which allow us to clarify the novelty of our work and its distinction from prior studies.

We agree with the Referee that the SOC in our proposed platform is similar to results previously reported in Ge-based heterostructures, including those in Ref. [50], and more notably Ref. [54], which employs a geometry more akin to ours. We have now explicitly acknowledged this in the main text, rather than only in the Appendix, and incorporated these references earlier in the manuscript.

However, we emphasize that the primary contribution of our work extends beyond simply demonstrating strong SOC. While SOC is an important ingredient, the novelty lies in the design of a realistic nanostructure that overcomes several critical challenges in the field and paves the way for experimentally feasible platforms for topological superconductivity. As we discuss in the Introduction, current full-shell hybrid NWs have significant limitations apart from small predicted SOCs, including the lack of control over the chemical potential and the proliferation of trivial subgap states (known as Caroli-de Gennes-Matricon analogs) with which potential Majorana zero modes have to coexist. Our proposed nanodevice addresses these challenges in several key ways. We propose to include an insulating InP core (absent in conventional full-shell NWs) that enables control over the Fermi level and confines the wavefunction to the outer GaSb shell, potentially reducing disorder. Moreover, in the presence of an outer SC shell, this confinement should enhance the induced superconductivity and, more importantly, should reduce the presence of trivial subgap states, as we discuss in the manuscript (see also the last answer below).

And we do so by leveraging the properties of the hole-bands of III-V SM compound heterostructures. While Ge-based (Group IV) heterostructures exhibit similar favorable SOC properties, they face several disadvantages compared to III-V compounds. Ge is more prone to dislocations and disorder due to its common growth methods and exhibits lower electron mobilities. Additionally, it is less versatile in tailoring band alignment, as III-V materials enable the use of ternary compounds to precisely engineer electronic properties. Growing a sharp, clean superconducting layer on Ge is also more challenging; its surface oxidizes quickly, requiring etching and preparation steps (leading to more disorder), and exhibits poorer chemical adhesion to SCs. Consequently, Ge-based SM-SC NW heterostructures are rarely studied, with most experiments focusing on planar nanostructures, which often suffer from irregular interfaces. This is why III-V SMs are widely studied and established in the field, particularly for partial-shell designs aimed at creating topological superconductivity. By identifying a III-V-based NW with strong hole-mediated SOC, our proposal provides an accessible and realistic alternative for many experimental groups that exclusively work in III-V platforms due to its advantages.

In summary, while the origin of the SOC that we report is not entirely new, the design and context of our nanodevice represent a significant step forward. Our proposal overcomes critical limitations of existing platforms, introduces a III-V-based alternative for realizing topological superconductivity, and opens up new opportunities for studying hole-mediated phenomena. As such, we believe our work fulfills SciPost Physics’ criterion of “opening a new pathway in an existing research direction, with clear potential for multi-pronged follow-up work.”

To address the Referee’s concern, we have revised the text to explicitly discuss the experimental challenges associated with existing platforms (with references), reinforcing the importance of finding an alternative platform outside Group IV materials. This clarifies what we initially meant by "specific design constraints," which, as we now realize, could be misinterpreted by a general reader. Furthermore, we have removed references to strain in Ge-based heterostructures, as we have verified that comparable SOC strengths can indeed be achieved in Ge without strain.

Other points with regards to this manuscript:

- I misunderstood the paper when I read it first. There is a reason for this: the authors use symbols as for the conduction band case ($\alpha$) and initially do not define what they mean with "spin-orbit" (this has only slightly improved in the revised version). In fact, Ref. [50] has the same term but does not call it spin-orbit interaction (Ref. [50] later derives a Rashba type SOI). I would suggest to the authors to (i) define what they call spin-orbit already in the beginning of Sec. 3.2, and not just in the appendix (and even with the appendix, I had to read all the previous papers to understand what was going on - it would be great to make this more self-contained), (ii) give an argument why they call this spin-orbit interaction.

We appreciate the Referee's thoughtful feedback and fully understand the confusion. In response to the Referee’s suggestion, we have now included an explicit definition of the SO interaction/coupling in the main text, specifically when we first introduce the parameter $\alpha$ in Section 3.2, in order to make the manuscript more self-contained and accessible to readers.

To completely clarify this point, we define the SO interaction, following the conventional definition, as an interaction that couples the momentum of a quasiparticle to its pseudo-spin. While the pseudo-spin in traditional cases typically refers to the electron's pure spin, in our work, it refers to a more complex band combination of spin-like degrees of freedom. Regardless of the precise nature of the pseudo-spin, the key point is that this interaction manifests in our case with a Rashba-type SO coupling, which is a scalar term that describes the strength of the coupling between the momentum and the pseudo-spin, i.e., the SO coupling $\alpha$. We hope this more precise definition resolves the confusion and makes the paper clearer for a broader audience.

- I find App. B a stronger argument for claiming that the effect is due to the Hamiltonian and not electric field than all the arguments in Sec. 3.2. Basically, Sec. 3.2 shows through numerics what does not affect the strength of the effect. App. B shows you get the effect from the Luttinger-Kohn Hamiltonian and confinement alone. To me, it would make more sense to state this fact first, and later show that the more detailed numerics only gives small corrections.

We appreciate the Referee’s valuable suggestion. We agree that the analytical derivation in Appendix B provides a stronger argument for attributing the effect to the bulk Hamiltonian structure together with the confinement, rather than to an electric field. However, this perception can depend on the reader's background field. We chose to present first the numerical results in Section 3.2, as they help to rule out other known sources of SO interactions before entering directly into a discussion of simplifications, approximations and notation. Nevertheless, we immediately after present the analytical calculations to justify our conclusions. Hence, since we believe that the Referee's proposed change does not significantly alter the overall understanding of our manuscript or the validity of our approach, we prefer to keep the current structure of our manuscript.

- Why do the authors consider an insulating core at all? It is clear from the results that the effect they want to show also will appear without an insulating core (as is the case in Ref. [50] for example).

It is important to understand that we are not just characterizing a hole-SM NW with strong SOC, we are proposing a specific SM nanostructure -the core-shell NW- with several crucial properties to improve the chances to find topological superconductivity in full-shell hybrid setups.

Although it is true that a strong SOC would be present without the insulating core (it actually slightly increases as the core diminishes for a fixed radius), the InP core serves additional purposes, as we discuss in Sections 2 and 4.

In particular, it helps to achieve the desired Fermi-level position. Notice that the Fermi level of GaSb does not naturally lie near the valence band. In contrast, the InP-GaSb heterostructure offers a type-II band alignment that places the Fermi level close to the GaSb valence band, as reported in Ref. [28]. Furthermore, the incorporation of dopants in the InP core can provide partial control over the Fermi-level position, without significantly affecting the electronic properties of the wire. This is important to achieve the proper electronic environment for topological phenomena.

In addition to this, the insulating core confines the wavefunction to the GaSb layer of thickness w, which then has a tubular shape. If the NW was then wrapped in a SC shell, the insulating core would help to enhance the proximity effect and, more importantly, in the presence of a threading flux, it should help to reduce the presence of trivial subgap states (known as Caroli-de Gennes-Matricon analogs) coexisting with potential Majorana zero modes and that are so detrimental for their topological protection, as discussed in Ref. 17.

---

## Round 2 · Author Response

Dear Editor,

We thank you and the Referees very much for the review of our manuscript. From the Referees input, we have made some changes to the manuscript (see below) that, although do not change the results and conclusions of our work, clarify several issues and significantly improve the quality of our work. We believe these changes satisfactory address all the questions/remarks the Referees had, and thus it is suitable for publication.

Best regards,
Samuel D. Escribano on behalf of all authors

---

## Round 2 · List of Changes

We refer to the responses to each referee to understand the motivation behind these changes. This is a comprehensive list with all the changes made:

- We have changed the title from "InP/GaSb core-shell nanowires: a practical proposal for Majorana modes in a full-shell hybrid geometry with hole bands" to "InP/GaSb core-shell nanowires: a novel hole-based platform with strong spin-orbit coupling for full-shell hybrid devices".

- We have modified/included two new sentences in the abstract. We no longer mention Caroli-de-Gennes subgap states, and we simply say "In addition, the SM wavefunction spreads all across the section of the nanowire, leading typically to a finite background of trivial subgap states with which MZMs may coexist". In addition, we include a better explanation of the SOC origin by saying "[...] that emerges from the combination of the intrinsic spin-orbit interaction of the SM active shell and the confinement effects of the nanostructure, thus depending...".

- In Section 1, at the end of the last paragraph, we have included the sentence: "We find that the SOC in these nanostructures originates from the combination of the intrinsic properties of the SM active layer and the radial confinement."

- In Section 2, in the fourth paragraph, we now include two more sentences. One is "However, different growing and fabrication methods, and particularly the ultimate incorporation of the SC outer shell in the hybrid NW, may change the position of this Fermi level. In this respect, we note that...". And the other one, at the end, is "Note also that other strategies to engineer the band alignment of hybrid quantum devices are currently being investigated. For example, in a recent experimental work [60], argon milling is used to modify the SC-SM interface while maintaining its high quality."

- In Section 2, at the end of the last paragraph, we clarify that "We make sure that our inhomogeneous FEM grid preserves the D6 symmetry of the hexagonal cross-section, which otherwise could introduce spurious solutions."

- In Section 3.2, in the last paragraph, we clarify that "Specifically, it is due to the radial confinement imposed by the nanostructure which breaks the translation symmetry of the crystal structure of this tetravalent SM".

- In Section 3.3, the third paragraph is new.

- In Section 3.4, the last paragraph has been extended.

- The title of Section 4 is now "Discussion and conclusions" and not only "Conclusions".

- In Section 4, in the first paragraph, we now clarify that "On the one hand, the tubular shape reduces the spread of the wave function across the NW section, confining it to the region close to the SC-SM interface. According to what happens in electron-based full-shell hybrid NWs, this should dramatically reduce the number of CdGM analogs coexisting with the MZMs [17]."

- In Section 4, in the second paragraph, we now explicitly say "Instead, it depends on the degree of HH and LH hybridization, ultimately regulated by the confinement strength provided by the NW radius."

- In Section 4, in the third paragraph, we include the sentence "Nonetheless, smaller radii require larger magnetic fields to achieve the topological phase, which may be detrimental to the parent superconductor. Therefore, a trade-off R must be chosen based on the SC material."

- In Section 4, the seventh and ninth paragraphs have been extended.

- In Section 4, we have included a new paragraph at the end.

- In the Appendix B, just before Eq. (39), we now clarify that "We note that this ansatz for the radial wavefunction is only valid for a core-shell NW with a thin w compared to R."

- Footnotes 5, 6 and 7 are new.

- Typos have been corrected, as well as some style issues.

- The references have been updated.

---

## Round 3 · Author Response

Thank you for giving us the opportunity to answer the last reports. We are also grateful to the Referees for their thoughtful feedback. We have revised the manuscript to address the remaining concerns raised by one of the Referees.
As you can see, two of the three Referees are satisfied with the previous changes and consider our work suitable for publication in SciPost Physics, highlighting its novelty and significance. The third Referee now acknowledges the correctness, clarity, and interest of our results, but remains unconvinced of the broader impact of our work. His/her primary concern relates to the similarity of the spin-orbit interaction (SOC) observed in our GaSb-InP nanodevice to that predicted for Ge-based nanowires.
While this observation is valid, we believe that the Referee (and perhaps also a general reader) may not be aware of the experimental challenges associated with Group IV nanowires, which limit their potential as a platform for realizing topological superconductivity. In contrast, our work introduces for the first time a nanowire design that takes advantage of the valence band properties (with strong SOC) of III-V semiconductor compounds. As many experimental groups exclusively work with III-V materials, particularly for the study of topological superconductivity precisely because of this reason, our proposal offers a realistic and accessible pathway for further exploration.
Furthermore, the significance of our work goes beyond the SOC strength. Our design incorporates an insulating core, which, although slightly diminishing the SOC, enables (i) wavefunction confinement close to the outer facet (where a potential superconductor would be placed), and (ii) partial tuning of the chemical potential — key requirements for achieving topological superconductivity.
To address these points, we have included an extended discussion comparing Group IV and III-V platforms to clarify the importance and implications of our proposal. Additionally, we have addressed all other minor comments from the Referee. As such, we believe that the manuscript clearly describes now its novelty and potential impact, and thus it should be suitable for publication in SciPost Physics.
Kind regards,
Samuel, on behalf of all the authors

---

## Round 3 · List of Changes

- References [27, 28] have been moved earlier in text (end of Section 1) and made clear that the SOC in our work is of the same kind than the one reported in these references. We have added a similar comment in the third paragraph of Section 3.3, when we describe the origin of the observed SOC.
- We have explained with more details the definition of SOC in our work at the end of the first paragraph in Section 3.2.
- We have added a more extended discussion about the experimental challenges of Ge-based heterostructures in the third paragraph of Section 2 (together with the footnote 3).
- We have removed all the comments about the fact that the SOC in Ge-based heterostructures depends on strain (as it is not completely true), including the abstract, Section 2, Section 4, and Appendix C.
- Several references have been added and updated.

---

## Editorial Decision

published